# Planar cell polarity coordination in a cnidarian embryo provides clues to animal body axis evolution

Julie Uveira[1], Antoine Donati[2], Marvin Léria[1], Marion Lechable[1], François Lahaye[1], Christine Vesque[2], Evelyn Houliston[1], Tsuyoshi Momose[1]*

[1]Sorbonne Université, CNRS, Laboratoire de Biologie du Développement de Villefranche-sur-mer (LBDV- UMR7009), Villefranche-sur-mer, France; [2]Sorbonne Université, CNRS UMR7622, INSERM U1156, Institut de Biologie ParisSeine (IBPS)- Developmental Biology Unit, Paris, France

## eLife Assessment

This analysis of the formation of the oral-aboral body axis in cnidarians, the sister group of bilaterians, is a significant and **fundamental** contribution to the field of Wnt signalling and planar cell polarity, particularly in or understanding in gradient formation, non-canonical Wnt signalling and Wnt-Frizzled interactions in cnidarians. The evidence supporting the conclusions is **compelling** and has the potential to contribute to a deeper understanding of the origin and evolution of Wnt signalling in cnidarians and metazoans in general. These findings, which are presented in a thoughtful and scholarly manner, will be of broad interest to developmental and evolutionary biologists.

**\*For correspondence:**
tsuyoshi.momose@imev-mer.fr

**Competing interest:** The authors declare that no competing interests exist.

**Abstract** Body axis specification is a crucial event in animal embryogenesis and was an essential evolutionary innovation for founding the animal kingdom. This process involves two distinct components that coordinate to establish the spatial organisation of the embryo: initiation of cascades of regionalised gene expression and orientation of morphogenetic processes such as body elongation. Intense interest in the first component has revealed Wnt/β-catenin signalling as ancestrally responsible for initiating regional gene expression, but the evolutionary origin of oriented morphogenesis has received little attention. Here, by addressing the cell and morphological basis of body axis development in embryos of the cnidarian *Clytia hemisphaerica*, we have uncovered a simple and likely ancestral coordination mechanism between Wnt/β-catenin signalling and directed morphogenesis. We show that the ligand Wnt3, known to initiate oral gene expression via localised Wnt/β-catenin pathway activation, also has a key β-catenin-independent role in globally orienting planar cell polarity (PCP) to direct morphogenesis along the oral-aboral axis. This PCP orientation occurs in two distinct steps: local orientation by Wnt3 and global propagation by conserved core PCP protein interactions along the body axis. From these findings, we propose novel scenarios for PCP-driven symmetry-breaking underlying the emergence of the animal body plan.

## Introduction

The earliest animals are thought to have been multicellular assemblages such as the 'choanoblastaea', which evolved from a unicellular ancestor (*Brunet and King, 2017*; *Maldonado, 2004*; *Nielsen, 2008*; *Sebé-Pedrós et al., 2017*). Subsequent evolutionary events that introduced a morphologically distinct primary body axis, i.e., breaking of tissue polarity symmetry, remain largely unaddressed (*Figure 1A*). In the animal lineage, Wnt/β-catenin-mediated specification of the primary body axis was likely an

**Figure 1.** Local Wnt3 globally orients the body axis tissue polarity and morphogenesis. (**A**) A key unanswered question of animal evolution is the origin of body axis patterning mechanisms, allowing initial multicellular assemblages to position specialised cell types (red) and coordinate cell polarity (represented in this diagram by oriented cilia and elongated shape). (**B**) Key stages of the early morphogenesis of *Clytia* embryos in relation to Wnt3 mRNA localisation (red). At the early gastrula (EG) stage, ectodermal cilia have formed, planar cell polarity (PCP) has developed along the future oral-aboral (OA)-axis, and the pointed morphology of the animal pole morphology can first be distinguished. Most axial elongation takes place during gastrulation. See also *Figure 1—figure supplement 1*. (**C**) Schematic representation of PCP in a single epidermal cell at the EG stage. Basal bodies are positioned on the oral side at the apical cortex (translational polarity). (**D**) Confocal image of the apical surface of a single epidermal cell stained for actin (cyan, phalloidin labelling) and γ-tubulin immunostaining (magenta), and a representation of the PCP vector, defined by the basal body position and its mirror image across the apical surface centroid as the initial and terminal points, respectively. (**E**) Local PCP coordination in control-MO- and Wnt3-MO-injected embryos at the EG stage. Bar: 10 μm. (Top) Confocal images as in D. (Bottom) PCP vector representation of the cells in the images. Dots represent cells with no clear polarisation or mitotic cells. Apical cell contours are wavy in Wnt3-MO embryos. (**F**) Distribution of cell orientation in individual scans (lines) and their average (histogram) for non-injected (373 cells, seven images), control-MO-injected (304 cells from three images) and Wnt3-MO-injected (225 cells from six images). The average orientation is defined as the origin (0°) for each scan. Horizontal bar: standard deviation. (**G**) The PCP polarisation index (degree of the polarisation of cells) by the length of the PCP vector, normalised by the cell size (****p<0.0001, Mann-Whitney U test). (**H**) Experimental procedures of the Wnt3-rescue experiment. Wnt3MO is injected into unfertilised eggs, then rescue mRNA solution with fluorescently labelled 10 kDa dextran is injected into a four-cell stage blastomere. (**I**) Circular colour-code representation of the PCP. (**J**) Wide-range PCP coordination observed in Wnt3-rescue experiments. Left: mock rescue control by water injection (N=9). Right: rescue by Wnt3 mRNA injection (N=27). Each set consists of three illustrations. Top: colour-wheel representation of the PCP. Cells derived from the Wnt3 or water-injected blastomere are at the top-right of each image, indicated with grey shading. Bar: 50 μm. Bottom-left: confocal thumbnail image. Cyan: actin; yellow: Dextran-Alexa Flour 647 (injected blastomere-derived cells). Bottom-right: radar plot of the cell orientation. The arrows in the radar show the circular mean of the PCP orientation within the single scan. circ s.d.: circular standard deviation. (**K**) Axial morphology of normal, Wnt3-MO-injected, and Wnt3-rescued embryos at late gastrula (LG) stage. Bar: 50 μm. Elongated body axis morphology was restored so that cells with injected Wnt3 mRNA were located at the induced oral (posterior). See also *Figure 3C*.

The online version of this article includes the following figure supplement(s) for figure 1:

**Figure supplement 1.** *Clytia* early embryogenesis stages and Wnt3 expression.

early innovation since it is found today across many species of both bilaterians (all of the animals' groups showing bilateral symmetry) and cnidarians (their sister group, comprising corals, sea anemones, jellyfish, etc.) (*Holstein et al., 2011*; *Loh et al., 2016*; *Petersen and Reddien, 2009*). Current evidence is insufficient to assess whether this innovation predated the branching of the more distantly related ctenophore and sponge clades (*Jager et al., 2013*; *Windsor Reid et al., 2018*). Importantly, however, transcriptional activation by β-catenin and its partner TCF at one pole cannot alone explain the morphological component of axial tissue patterning (*Philipp et al., 2009*). Rather, polarity information capable of directing axis development can be distributed globally across the whole embryo, as clearly illustrated by classic experimental manipulations in the hydrozoan *Clytia gregaria*, which led to the concept of global polarity (*Freeman, 1981*; *Primus and Freeman, 2004*).

In retrospect, Freeman's global polarity concept has much in common with planar cell polarity (PCP), the mechanism coordinating the cell polarity in the plane of epithelial tissues, later identified and unravelled genetically in *Drosophila* and vertebrates. PCP involves interactions between neighbouring cells, with each cell in a sheet, thus carrying polarity information. It is mediated by interactions of core PCP proteins: the Wnt receptor Frizzled (Fz), intracellular signal transducer Dishevelled (Dsh), transmembrane protein Strabismus/Van Gogh (Stbm), LIM-domain protein Prickle (Pk), and G-protein-coupled receptor cadherin Flamingo (Fmi), to ensure tissue polarity and enable morphogenesis (*Butler and Wallingford, 2017*; *Gray et al., 2011*; *May-Simera and Kelley, 2012*; *Seifert and Mlodzik, 2007*; *Vladar et al., 2009*; *Zallen, 2007*). A conserved feature of PCP is the polarised localisation of the Stbm-Pk-Fmi complex on one side of the apical cortex of each cell and the Fz-Dsh-Fmi complex on the other side. These two complexes interact between adjacent cells to coordinate PCP orientation of neighbouring cells. The PCP proteins are conserved across bilaterians and cnidarians, as well as in some sponge groups (*Hale and Strutt, 2015*; *Lapébie et al., 2011*; *Schenkelaars et al., 2016*). Fz and Dsh are components both for core PCP and Wnt/β-catenin signalling, so Wnt, Fz and Dsh are potentially key elements in coupling the pathways.

Two crucial issues for the developing embryo and the earliest animal ancestors are (i) to orient PCP and thus morphogenesis consistently and globally across the whole embryo and (ii) to coordinate PCP orientation with regionalised cell fate specification initiated by Wnt/β-catenin signalling. These issues have not been addressed in the context of the primary body axis specification, while current examples of PCP orienting mechanisms from other tissues and animals show considerable variability (*Butler and Wallingford, 2017*).

We address here the coupling of Wnt signalling and PCP during body axis specification in a cnidarian in order to shed light on the roles of PCP in the evolutionary emergence of the metazoan body axis, exploiting the jellyfish model species *C. hemisphaerica* (*Houliston et al., 2022*). Previously, we showed that maternal mRNAs for the ligand CheWnt3 (orthologue of bilaterian Wnt3, hereafter called Wnt3) and its receptors are localised with respect to the animal-vegetal polarity of the unfertilised *Clytia* egg (*Figure 1B*), and activate Wnt/β-catenin signalling in a future oral (posterior) domain of the developing embryo to control gene expression along the developing primary body axis (oral-aboral [OA]) (*Momose et al., 2008*; *Momose and Houliston, 2007*). In parallel, conserved core PCP proteins, including Stbm orthologue CheStbm (Stbm), are responsible for establishing tissue polarity (PCP) in the epidermis to support directed larval swimming via cilia beating along the OA-axis and are also required for elongation along the primary body axis during gastrulation (*Momose et al., 2012*). Stbm protein is localised on the aboral side of each cell (*Momose et al., 2012*) as expected from *Drosophila* and vertebrate PCP deployment mechanisms.

In the current work, we explore the hypothesis that Wnt3, as well as activating Wnt/β-catenin signalling (*Momose et al., 2008*), provides the initial cue to direct global orientation of PCP and thereby define the *Clytia* embryonic body axis both molecularly and morphologically. A first hint of this from our previous work was that the characteristic and complete absence of embryonic axis elongation observed following Wnt3 knockdown by Morpholino (Wnt3-MO) injection could be restored by injection of synthetic Wnt3 mRNA into single blastomeres during early development (*Momose et al., 2008*). Importantly, Wnt3 is the sole Wnt mRNA detected before the early gastrula (EG) stage, the period when axial morphology and epidermal PCP are established, with other Wnt genes being expressed downstream of Wnt3 or at subsequent stages of larval development (*Lapébie et al., 2014*; *Momose et al., 2008*; *Figure 1B*, *Figure 1—figure supplement 1*). This greatly facilitates the assessment of Wnt ligand function in directing PCP, in contrast to traditional model species where multiple

Wnt ligands have parallel roles during embryogenesis. The studies reported here reveal how local Wnt3 cues orient the global axial PCP and PCP-driven oriented morphogenesis in the *Clytia* embryo. Two discrete steps define the PCP orientation: a local cueing step with orally localised Wnt3 signal and a global propagation and coordination step mediated by core PCP proteins. Interestingly, while the role of Wnt3 is the instructive cue for PCP orientation and thus morphological axis development during *Clytia* normal development, other cues, including mechanical strains, can orient PCP. A similar variety of PCP orientation cues has been reported in *Drosophila* and vertebrates. Our findings allow us to propose a 'PCP-first' model for body axis symmetry-breaking in the metazoan ancestors, in which the self-organising activity of the PCP pathway allowed body axis innovation in early animal ancestors.

## Results

### A local Wnt3 cue orients global PCP

We first analysed in detail PCP in Wnt3-depleted embryos at the EG stage, the earliest stage that shows morphological OA polarity (*Figure 1B*, *Figure 1—figure supplement 1*). PCP manifests as structural and positional asymmetries of the ciliary basal bodies, which are consistently positioned towards the oral side of the apical cortex (*Figure 1C*). As previously reported (*Momose et al., 2012*), we quantified epidermal PCP orientation by assessing the basal body positioning (*Figure 1D*). In unmanipulated or control-MO-injected embryos, cells showed coordinated PCP, with 85% of cells orienting within ±45° (s.d. 33° and 34°, respectively) of the average orientation. In embryos injected with Morpholino antisense oligonucleotides (Wnt3-MO) to block Wnt3 translation, PCP was oriented nearly completely randomly (*Figure 1E and F*) (s.d. 84°, comparable to 95° for Stbm-MO embryos, *Momose et al., 2012*), and individual cells were less polarised (*Figure 1G*), reflecting the complete lack of axial morphology. The apical cell boundary was wrinkled in Wnt3-MO-injected cells (*Figure 1E*), as observed following Fz1-MO, Dsh-MO, or Stbm-MO injections (*Momose et al., 2012*), indicative of markedly reduced cortical tension. These analyses demonstrate that Wnt3 depletion causes PCP defects equivalent to the knockdown of the core PCP proteins (*Momose et al., 2012*). This supports the hypothesis that Wnt3 provides a local cue to orient global PCP, which is responsible for the development of cilia polarity and axially elongated morphogenesis in *Clytia*.

To test this hypothesis, we injected 100 ng/µl Wnt3 mRNA and fluorescently labelled Dextran into one blastomere at the four-cell stage in Wnt3-MO embryos, which we call the 'Wnt3-rescue' experiment (*Figure 1H and J*). At 10 ng/µl Wnt3 mRNA, axis restoration was random, and either quasi-complete axis restoration or no axis formation was observed. At the EG stage, we quantified PCP in a broad area (246×246 µm$^2$) and represented it by bars with a circular colour code (*Figure 1I*). Control mock injections did not affect the PCP defects caused by Wnt3-MO. In contrast, coordinated PCP was restored by Wnt3-rescue in a non-autonomous manner (*Figure 1J*). Restored ciliary alignment was observed at least 200 µm away from the Wnt3 mRNA-positive area. The Wnt3 source likely influences PCP at even further distances as a virtually complete morphological axis was restored by the late gastrula stage, with the Wnt3-mRNA lineage forming a new oral pole (*Figure 1K*) as shown previously (*Momose et al., 2008*). Locally expressed Wnt3 signal is thus necessary and sufficient to orient the PCP along the whole body axis.

### Rapid global PCP development precedes basal body alignment

To understand the initial process by which PCP becomes coordinated by Wnt3, we examined stages preceding EG, again using basal body localisation as the read-out. We reasoned that if PCP propagation occurs through successive cell interactions, cell alignment might spread as a wave across the embryo. PCP was detectable from the mid-blastula stage (8–9 hpf, *Figure 2A–D*), slightly after the onset of the zygotic Wnt3 expression (*Momose et al., 2008*) and epithelialisation of the blastula cells (*Kraus et al., 2020*). PCP was slightly better coordinated in the region proximal to the Wnt3-positive area (*Figure 2C and D*). We did not, however, detect any wavefront of local cell alignment; rather, the PCP gradually increased in coherence across the embryo over time (*Figure 2A and B*). This was confirmed by live imaging, injecting synthetic mRNAs in the egg to visualise cell contours with PH-Venus and basal bodies with Poc1-mCherry (*Figure 2E*, *Video 1*). The movies further revealed active mitosis in the blastodermal cells. A weak bias in cleavage orientation was observed along the OA-axis (*Figure 2F*). During mitosis, duplicated centrioles from the ciliary basal bodies are deployed

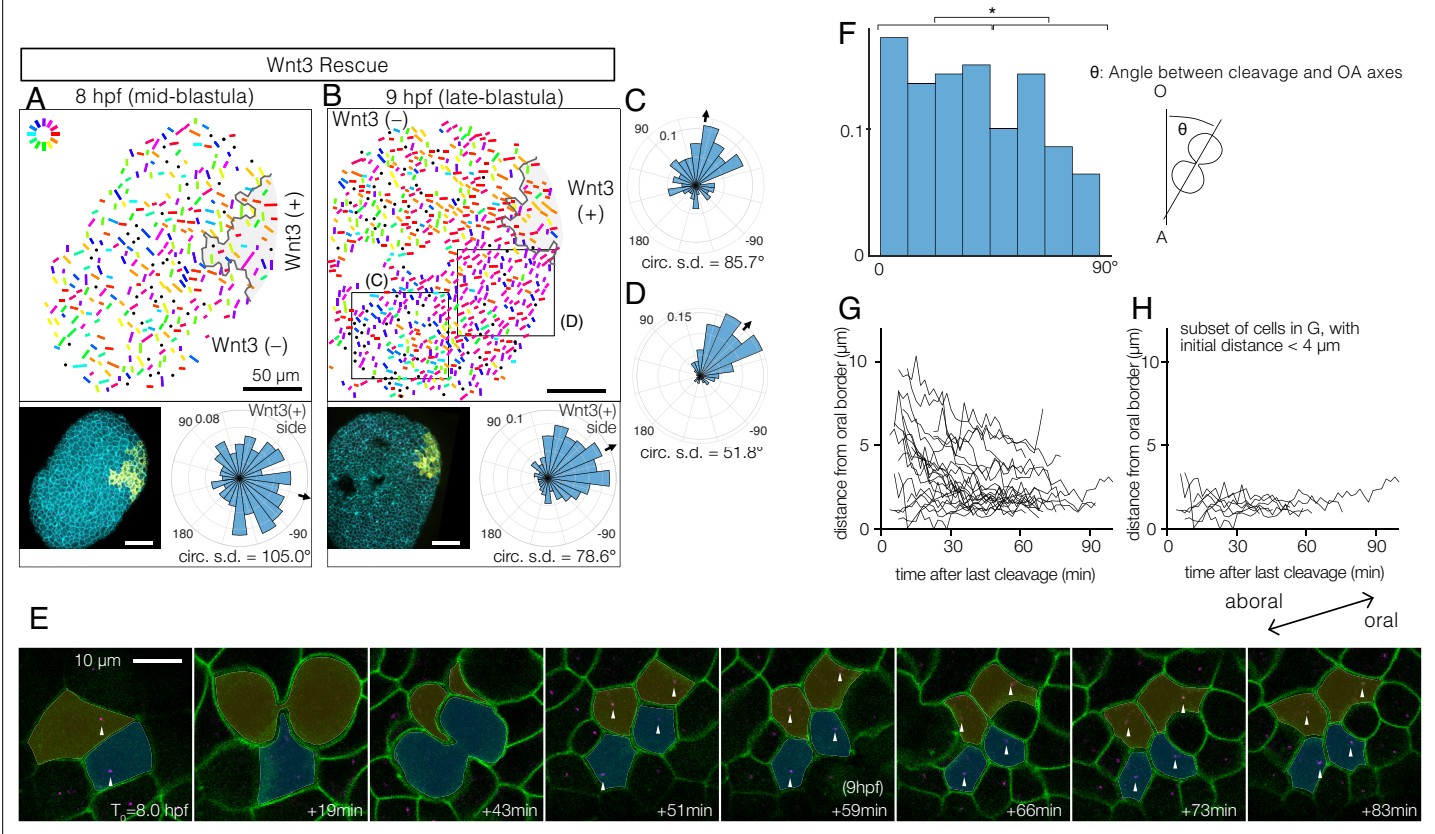

**Figure 2.** Wnt3-driven planar cell polarity (PCP) orientation precedes visible cell alignment. (**A, B**) PCP orientation in Wnt3-rescue embryo at the mid-blastula stage (A: 8 hpf, N=10) and late-blastula stage (B: 9 hpf, B, N=10). Graphical PCP representations and radar histogram plots are as in *Figure 1*. (**C, D**) Radar plot of PCP in distal (**C**) and proximal (**D**) areas of the region shown in (**B**). (**E**) Basal body displacements after cell divisions. Bar: 10 µm. Cell membranes labelled by PH-Venus (green) and basal bodies with Poc1-mCherry (magenta). Arrows indicate basal bodies positioned on opposite sides of cleavage planes located after being engaged to the spindles. Aborally displaced basal bodies migrate towards the oral side. The average PCP defined the oral-aboral (OA)-axis. (**F**) Distribution of the angles between the cleavage orientation axis to the OA-axis defined by the PCP orientation (0–90°, n=131). Cleavages along the OA-axis (0–45°) are favoured (*p<0.05 chi-square test). (**G**) Directional migration of the basal body towards the oral cell boundary after cell cleavage. Each line indicates the distance of a single basal body to the oral edge of the cell after the last cleavage. Basal bodies migrate towards the oral cell boundary, suggesting PCP has been established by the time PCP becomes structurally manifest. (**H**) Subset of (**G**) where the initial distance to the oral edge is less than 4 µm after the cell division.

in the mitotic spindles (*Figure 2E*, *Video 1*) such that basal body positioning becomes scrambled immediately after mitosis. Subsequently, the displaced basal bodies migrate unidirectionally towards the oral side of each cell, where the cilium is re-established, a clear indication that PCP was already in place across the tissue (*Figure 2E, G, and H*). Together, these observations indicate that PCP is established at the mid-blastula stage, presumably via canonical localisation of core PCP proteins to opposite cortices of each cell through continuous Fz-Stbm interaction. The protein partitioning then manifests as asymmetric basal body positioning. Localised PCP proteins may attract basal bodies to the oral cortex of the cell as described in ascidian embryos (*Negishi et al., 2016*). During subsequent stages, PCP is either maintained (*Bellaïche et al., 2004*) in each cell during mitosis or rapidly restored through interactions with neighbouring cells (*Devenport et al., 2011*).

## Wnt3 orients PCP in a β-catenin-independent manner

We previously showed that local Wnt3 expression is necessary and sufficient to activate the Wnt/β-catenin pathway in the oral pole of the *Clytia* embryo (*Momose et al., 2008*). We next examined whether the action of Wnt3 for PCP orientation and morphogenesis is mediated by the Wnt/β-catenin pathway. First, we showed that PCP orientation by Wnt3 is independent of β-catenin: ectopic Wnt3 could orient PCP in the Wnt3-rescue experiment even in the presence of a dominant-negative form of CheTCF

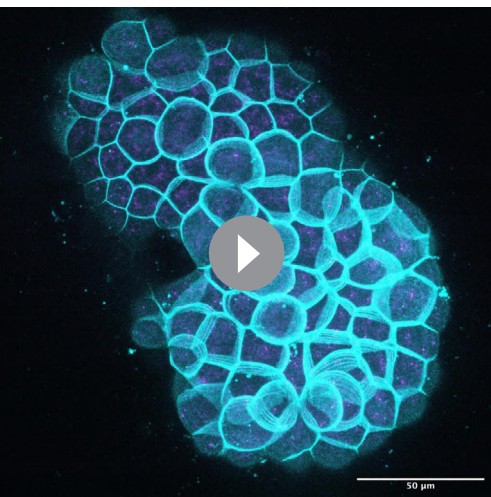

**Video 1.** Live imaging of basal body migration at the blastula stage. (193 μm x193 μm field of view, 570 times speed).

https://elifesciences.org/articles/104508/figures#video1

(dnTCF) (*Figure 3A*), which blocks downstream transcription of the β-catenin pathway. Gastrulation was completely prevented, confirming that the Wnt/β-catenin pathway was inactive (*Figure 3C, D*, *Figure 3—figure supplement 1*). Furthermore, body axis elongation was partially restored at the EG stage in the absence of an active β-catenin pathway (*Figure 3C*), although they did not develop the pointed morphology of the oral pole nor elongate further (*Figure 3D*, *Figure 3—figure supplement 1*). These results support a model in which axial elongation requires both PCP coordination and gastrulation (migration and intercalation of ingressed endodermal cells), as suggested previously (*Kraus et al., 2020*; *van der Sande et al., 2020*). Conversely, we found that constitutively active β-catenin (CA-β-catenin) mRNA was not able to restore PCP coordination nor elongation of Wnt3-depleted embryos (*Figure 3B*) but did restore β-catenin-dependent gastrulation (*Figure 3—figure supplement 1*).

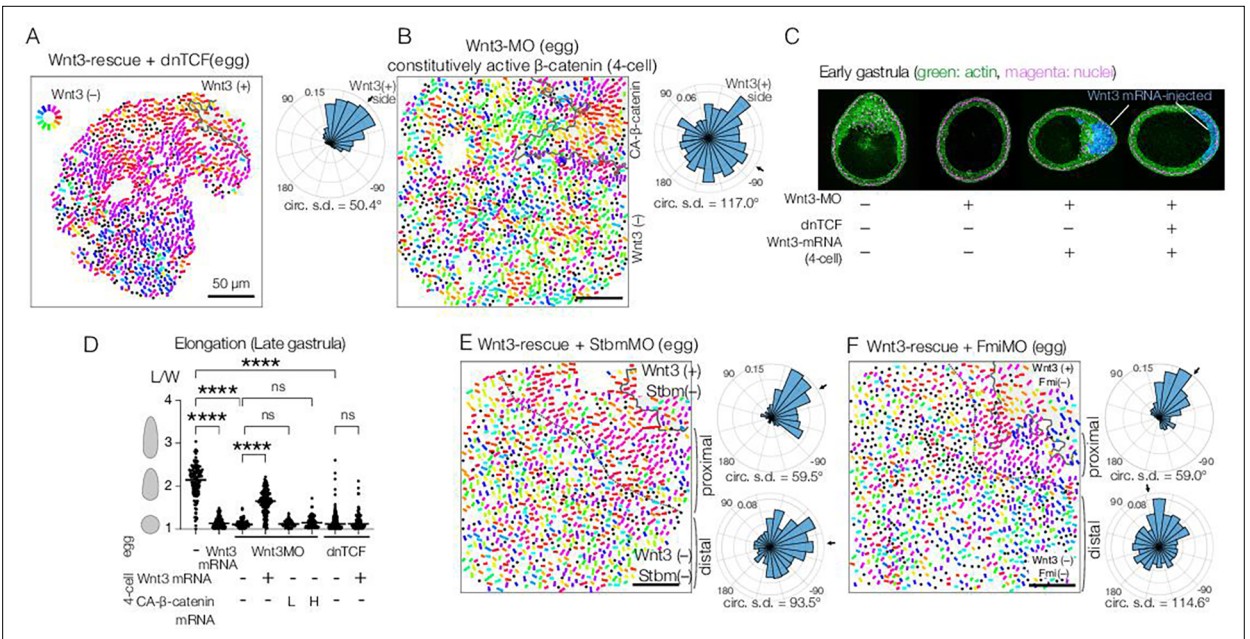

**Figure 3.** Planar cell polarity (PCP) orientation by Wnt3 is Wnt/β-catenin-independent, and its propagation across the body axis requires Stbm and Fmi. (**A**) Wnt3-rescue with an additional injection of a dominant-negative form of TCF (dnTCF) into the egg, N=10. (**B**) Wnt3-rescue by the constitutively active form of β-catenin (CA-β-cat) instead of Wnt3 mRNA, N=7. (**C**) Onset of axial morphology in early gastrula (EG) embryos. Localised Wnt3 is necessary and sufficient for the onset of the elongation towards its source. Green: actin (phalloidin), magenta: nuclei (To-Pro3), and blue: Wnt3-mRNA/Dextran-injected cell lineage. (**D**) Roles of Wnt3 and β-catenin in the axial elongation in late gastrula (LG) embryos. The elongation was measured by the elongation index (L/W): the primary axis length (**L**) divided by its perpendicular length (**W**), ****p<0.0001, Mann-Whitney U test. Both local Wnt3 expression and β-catenin/TCF-dependent gastrulation were required for axial elongation. See *Figure 3—figure supplement 1*. (**E, F**) Wnt3-rescue experiments with additional injections of (**E**) Stbm-MO (N=23) and (**F**) Fmi-MO (N=6). Radar plots of the PCP orientation show the manually defined proximal area (proximal: between the dotted line and the Wnt3-positive region) and distal area (distal: opposite area to the Wnt3-positive region across the dotted line). PCP is correctly polarised without Stbm or Fmi close to the Wnt3-injected cell clone but not in the distant area.

The online version of this article includes the following figure supplement(s) for figure 3:

**Figure supplement 1.** CheWnt3 is necessary and sufficient to orient the morphological oral-aboral (OA)-body axis, while the Wnt/β-catenin pathway acting under CheWnt3 is necessary but not sufficient.

Taken together, these experiments demonstrate that global PCP orientation along the *Clytia* OA-axis by Wnt3 is not mediated by the Wnt/β-catenin pathway. Wnt3 thus has dual independent roles: PCP orientation and Wnt/β-catenin activation.

## Global PCP is established in two distinct steps

We next tested the contribution of the core PCP proteins CheStbm and CheFmi (hereby called Stbm and Fmi) to linking PCP orientation to the local Wnt3 source. As expected, locally confined expression of Wnt3 mRNA did not restore global PCP coordination in Stbm-MO or Fmi-MO embryos (*Figure 3E and F*). Importantly, however, polarity was restored in the area immediately adjacent to the Wnt3 mRNA-positive cell patch. PCP orientation seemed less coordinated within the Wnt3-mRNA-positive area. The effect covered a distance of a few cells, potentially derived from cells directly in touch with the Wnt source at earlier stages. This observation raised the possibility that global PCP orientation in *Clytia* embryos is established in two steps: Stbm/Fmi-independent PCP orientation by local gradient or contrast of Wnt3 activity at the oral end, followed by global PCP propagation through the intercellular interaction of core PCP proteins (two-step model), as predicted by the domineering non-autonomy property of mosaic PCP mutations (*Amonlirdviman et al., 2005*). Alternatively, Wnt3 may diffuse from the oral Wnt3-expressing cells (*Figure 1A*) to form a long-range gradient along the OA-axis and orient the PCP (Wnt-gradient model). Wnt activity gradients have been shown to control or orient PCP in mice (*Gao et al., 2011*; *Minegishi et al., 2017*).

To distinguish these two models, we created mosaic embryos with a small patch of core-PCP negative cells within Wnt3-rescued embryos.

To achieve this, we transplanted single 32- or 64-cell stage blastomeres from Stbm-MO/Wnt3-MO- or Fmi-MO/Wnt3-MO-injected embryos to Wnt3-rescued embryos with Dextran lineage markers (*Figure 4A–E*). The two-step model predicts that the PCP orientation would make a detour around the Stbm-MO/Fmi-MO mosaic area; in contrast, under the Wnt-gradient model, PCP would be affected only in the mosaic area. We found that in the Stbm-MO or Fmi-MO mosaic Wnt3-rescued embryos, PCP was correctly oriented or slightly disturbed within the Stbm-MO patch while often de-polarised in the Fmi-MO patch. Outside the Sbtm-MO or Fmi-MO patches, PCP orientation was inverted on the aboral side of the patches unless these were in close proximity to the Wnt3-mRNA patches. Stbm- or Fmi-defected cells thus exhibited domineering non-autonomy effects. This is consistent with the intercellular Fz-Fmi and Sbm-Fmi interaction model shown in *Drosophila* and vertebrates and the localisation of Stbm protein on the aboral side of each epidermal cell in *Clytia* (*Momose et al., 2012*). PCP was oriented normally in the cells on the oral or lateral sides of the patches (*Figure 4B and D*). PCP orientation from oral to aboral detouring around the Stbm-negative patches was similarly observed when mosaics were generated by injection of Stbm-MO into a single blastomere in a Wnt3-rescue embryo at the 16-cell stage (*Figure 4F and G*). These observations strongly support the two-step model in which Wnt3 acts locally to orient PCP, and PCP coordination is then transmitted from oral (i.e. Wnt3-positive lineage) to aboral through core PCP interactions (*Figure 4H*). They are not compatible with the alternative model that a long-range Wnt3 gradient orients PCP.

## Mechanical cues can orient PCP

Transplantation of Wnt3-MO blastomeres into Wnt3-rescued host embryos at the 32- or 64-cell stage did not cause any PCP defects in most cases. However, in a few cases (N=3 out of 8), reorientation of PCP towards the transplanted donor lineage was observed (*Figure 5A–E*). In these cases, the donor cells were incompletely incorporated into the host ectodermal epithelium (*Figure 5D*), such that the donor cells remained as a protrusion with a 'rosette' (radial) arrangement. In these embryos, host and donor-derived epithelial cells acquired an elongated morphology around the circumference of the smooth host-donor boundary (*Figure 5B*). Although the grafted tissue influenced the orientation of PCP in the surrounding host cells, the direction was opposite to that driven by Wnt3-expressing sources. The rosette-like transplants thus operate as artificial 'aboral' cues, in contrast with Wnt3 sources, which provide an oral cue (*Figure 5C*). Unlike Stbm-MO or Fmi-MO mosaic embryos, cell polarity was continuous across the host-transplant border in these Wnt3-MO to Wnt3-MO transplantations (*Figure 5A*), suggesting the core PCP protein interaction was not disturbed at the boundary and thus not likely due to the non-autonomous effect of PCP defects. Epithelial 'scarring' can thus act to orient PCP, perhaps via discontinuities in cell adhesion or mechanical strain. PCP reorientation by

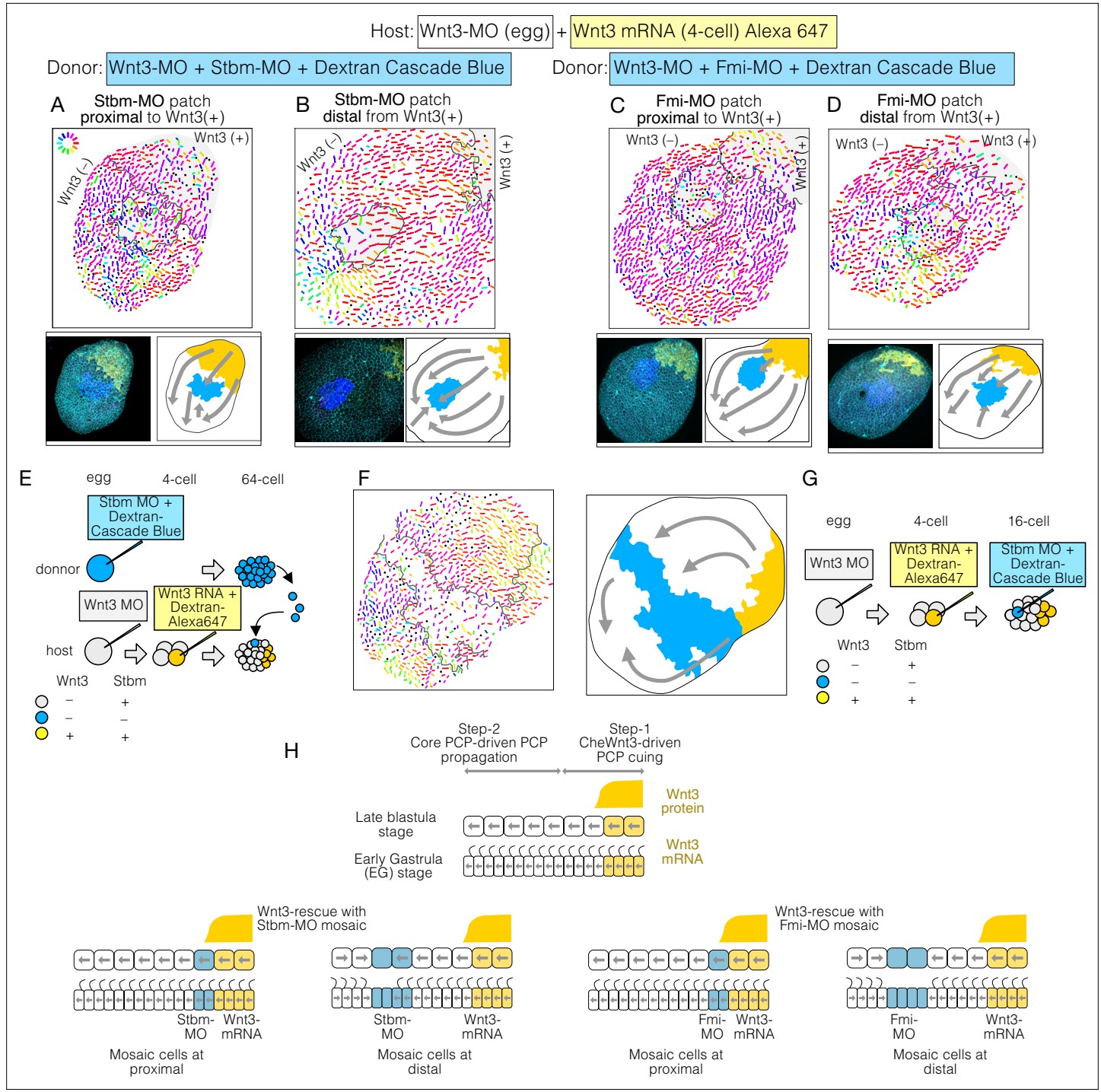

**Figure 4.** Planar cell polarity (PCP) behaviour in Stbm/Fmi-mosaic/ Wnt3-rescue experiments supports the two-step model for global PCP orientation. (**A–D**) Wnt3-rescued early gastrula embryos, including Stbm(-) or Fmi(-) mosaic patches made by blastomere transplantation. (**A, B**) Blastomeres from Wnt3-MO/Stbm-MO double-injected embryos were transplanted as the donor into Wnt3-rescued host embryos at the 64-cell stage. They happened to be incorporated in close/proximal (A: 658 cells, N=4) or far/distal (B: 757 cells, N=4) positions with respect to the Wnt3 source, respectively. (**C, D**) The same experiment was conducted with Wnt3-MO/Fmi-MO-injected blastomeres as the donor, incorporated in proximal (C: 984 cells, N=3) and distal (D: 649 cells, N=3) positions, respectively. (**E**) Schematic drawing of the experimental procedure of the transplantation. (**F**) Wnt3-rescued early gastrula embryos with Stbm(-) mosaic patch generated by additional injection of Stbm-MO at the 16-cell stage (N=1). (**G**) Schematic drawing of the experimental procedures for (**F**). The flow of the PCP orientation is indicated by grey arrows in the bottom right drawings in (**A–D**) and right in (**G**). (**H**) Graphical summary of Stbm-MO and Fmi-MO mosaic experiments. The graphical representation and thumbnail confocal images indicate Wnt3-positive lineages and Stbm-MO- or Fmi-MO-injected lineages in yellow and blue. All vector representations are as in *Figure 1*.

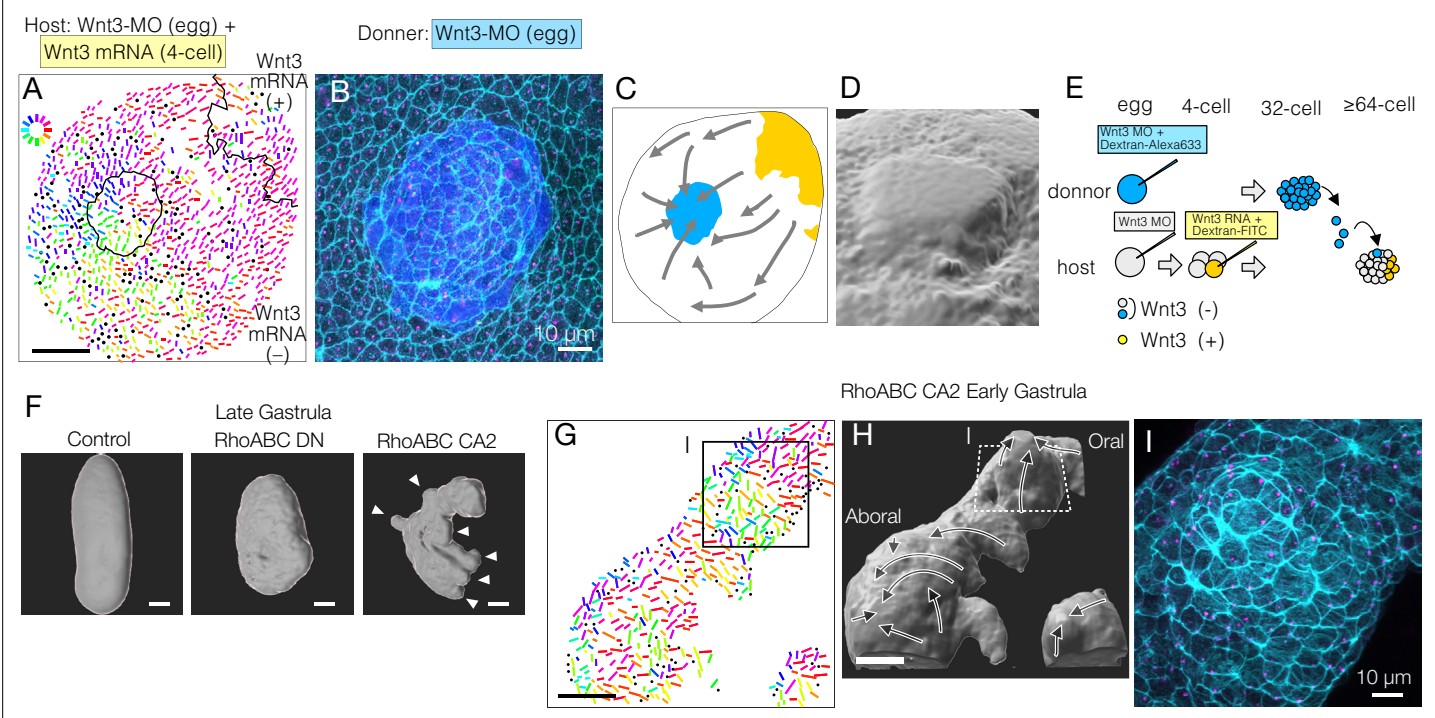

**Figure 5.** Mechanical strains can reorient planar cell polarity (PCP). (**A–E**) PCP orientation induced by transplantation from a Wnt3-MO donor into the Wnt3-MO background of a Wnt3-rescue host. (**A**) Incomplete incorporation of host and donor cells into a smooth epidermis (N=3/8) causing long-range PCP orientation, as if the donor acts as an aboral cue, in addition to the Wnt3 oral cue. (**A**) PCP representation by colour-coded bars. See also the legends for *Figure 1*. (**B**) Confocal image of the apical cortex of the epidermis showing the rosette structure caused by incomplete incorporation of the donor cells. Both host and donor cells are elongated around the graft boundary. (**C**) Graphical summary of the PCP orientation: Wnt3-mRNA lineage in Wnt3-rescue host in yellow, Wnt3-MO graft in blue. (**D**) 3D reconstruction of the rosette structure from the confocal images by contouring cortical actin signals. (**E**) Schematic representation of the transplant experiment. (**F**) 3D representation of the late gastrula morphologies induced by mRNA injection of dominant-negative form (RhoABC-DN) and constitutively active forms (RhoABC-CA2) of the PCP effector RhoABC, which caused reduced elongation and extra protrusions, respectively. The overall oral-aboral (OA)-axis remained distinguishable based on morphology and gastrulation. (**G–I**) PCP is coordinated consistently with respect to the induced protrusions caused by RhoABC-CA2. (**G**) Colour-wheel representation of PCP in the RhoABC-CA2 injected early gastrula embryos; the presumed oral pole is to the top-right. (**H**) 3D reconstruction of the tissue morphology for the same specimen, with PCP flow represented by arrows. (**I**) Confocal image of the apical cortex of the epidermis at the induced protrusion, corresponding rectangles in G and H. Cortical actin fibres are circumferentially organised around the protrusion. Bars are 50 µm except for 10 µm in B and I. See *Figure 1* for the circular colour-code representation of PCP orientation. Cyan: actin; magenta: γ-tubulin; blue: donor lineage marker Dextran for B and I.

mechanical strain has been reported in wing PCP repatterning in *Drosophila* and for the orientation of *Xenopus* epidermal motile cilia (*Aigouy et al., 2010*; *Chien et al., 2022*; *Guirao et al., 2010*; *Mitchell et al., 2007*; *Olguín et al., 2011*). We also performed transplantations into Wnt3-MO-only hosts but did not obtain any such rosettes, so we could not determine whether Wnt has a permissive role for PCP in this context. Nevertheless, we can conclude that any local PCP asymmetry can potentially act as a PCP orientation cue via core-PCP proteins, and thus again is consistent with the two-step orientation model.

To explore further whether mechanical strain can orient PCP, we disrupted the morphology of the embryo by expressing dominant-negative and constitutively active forms of *Clytia* RhoABC, the single cnidarian orthologue of the vertebrate Rho GTPases, including RhoA. Late gastrula embryos expressing dominant-negative RhoABC showed defects in axial elongation, consistent with a role in morphogenesis. Conversely, embryos injected with constitutively active RhoABC showed a scrambled axial morphology with ectopic protrusions induced along the major axis (*Figure 5F*). The induced protrusions appeared prior to gastrulation. PCP orientation was locally coordinated along the microaxes of each protrusion so that they became extra aboral poles with respect to PCP (*Figure 5G and H*), while PCP orientation along the major body axis was globally maintained. Enhanced cortical actin fibres were arranged around the circumference of the induced protrusion (*Figure 5I*). This method to provoke mechanical deformation of the embryo thus generated protrusions that mimicked the

morphological extension and circumferential cell arrangement seen in the transplantation-induced rosettes, with in both cases PCP orientation aligning along the experimentally formed ectopic mini-axes.

## Discussion

In this study, we have shown that during embryogenesis in the cnidarian *Clytia,* the ligand Wnt3, known to initiate oral gene expression via Wnt/β-catenin pathway activation, also provides β-catenin-independent cues to orient PCP globally in the embryo and thereby direct morphogenesis along the OA-axis. PCP orientation occurs in two distinct steps: local orientation by Wnt3 and global propagation via conserved core PCP machinery. These findings demonstrate deep conservation of PCP orientation and tissue morphogenesis mechanisms in eumetazoans and allow us to propose novel scenarios for PCP-driven axis symmetry-breaking underlying the emergence of the animal body plan.

### A single Wnt ligand coordinates molecular and morphological axis determination

Wnt/β-catenin signalling (*Holstein et al., 2011*; *Loh et al., 2016*; *Petersen and Reddien, 2009*) and PCP pathway components (*Gray et al., 2011*; *Kumburegama et al., 2011*; *Momose et al., 2012*; *Wallingford, 2012*; *Zallen, 2007*) are involved respectively in embryonic body axis specification and morphogenesis across many metazoan species, but how these aspects are linked has been complicated to address. For instance, in vertebrates, Wnt5a and Wnt11 play crucial roles during body axis morphogenesis (*Heisenberg et al., 2000*; *Kilian et al., 2003*; *Tada and Smith, 2000*; *Wallingford and Harland, 2001*), but it is not straightforward to demonstrate whether they provide positional cues to orient PCP along the body axis, while they do in other contexts (e.g. orientation of ciliated node cells and muscle fibre) (*Gros et al., 2009*; *Minegishi et al., 2017*). During *Clytia* embryogenesis, a single Wnt ligand directs both molecular and morphological aspects of axis formation. Wnt3 translated from maternally localised mRNA controls axial gene expression along the body axis via β-catenin regulation (*Momose et al., 2008*). This dual role of Wnt3 elegantly ensures the coherence of gene expression along the developing OA-axis with morphogenetic aspects such as embryo elongation during gastrulation and beating of ectodermal cilia for directed locomotion. Maternal XWnt11 in *Xenopus* activates the Wnt/β-catenin pathway (*Tao et al., 2005*), and it is also implicated in the β-catenin-independent role to control convergent extension (*Tada and Smith, 2000*). The dual roles of a Wnt ligand for PCP and Wnt/β-catenin pathway may be a common strategy in vertebrates and cnidarians.

### Two-step PCP orientation, initiated by CheWnt3 and propagated by core PCP interactions

The experiments presented here reveal two distinct steps of Wnt3-driven PCP and axial morphology in *Clytia* embryos (*Figure 6A*): A Wnt3-dependent local PCP orientation (cueing) step is followed by a propagation step mediated by core PCP proteins. This resembles the 'domineering non-autonomy' phenomenon described in *Drosophila* mosaic mutants for core PCP genes (*Amonlirdviman et al., 2005*). How Wnt3 locally orients PCP in the cueing step in *Clytia* embryos remains to be understood. Locally restricted PCP orientation by Wnt3 in the presence of Stbm-MO or Fmi-MO suggests a local gradient, or contrast, of Wnt3 acts as the orientation cue. Indeed, axis polarity is lost under uniform expression of Wnt3 (*Momose et al., 2008*). A likely candidate mediating Wnt3 cue is CheFz1 (a single orthologue to vertebrate Fzd1/2/7/3/6), whose predicted localisation is on the oral side of each cell, opposite to the aboral accumulation of Stbm. The role of CheFz1 for mediating PCP orientation remains to be revealed, despite the effect of CheFz1 knockdown in PCP coordination (*Momose et al., 2012*). Other Wnt3 receptor candidates are CheFz2 (Fzd5/Fzd8) and CheFz4 (Fzd4). An attractive hypothesis is that the local Wnt3 gradient, or contrast, anchors one of the Fz receptors at the oral side of the cells adjacent to the Wnt3-positive area in a mechanism similar to zebrafish Wnt11 that locally accumulates Fz7 in the plasma membrane (*Witzel et al., 2006*). A further implication of the two-step model and the domineering non-autonomy character of PCP proteins is that PCP orientation may act as a morphogen-independent body axis maintenance mechanism in certain regeneration contexts in cnidarians (*Livshits et al., 2017*; *Sinigaglia et al., 2020*). Established local PCP coordination may be propagated to restore the body axis in regeneration. In this context, it would be interesting to

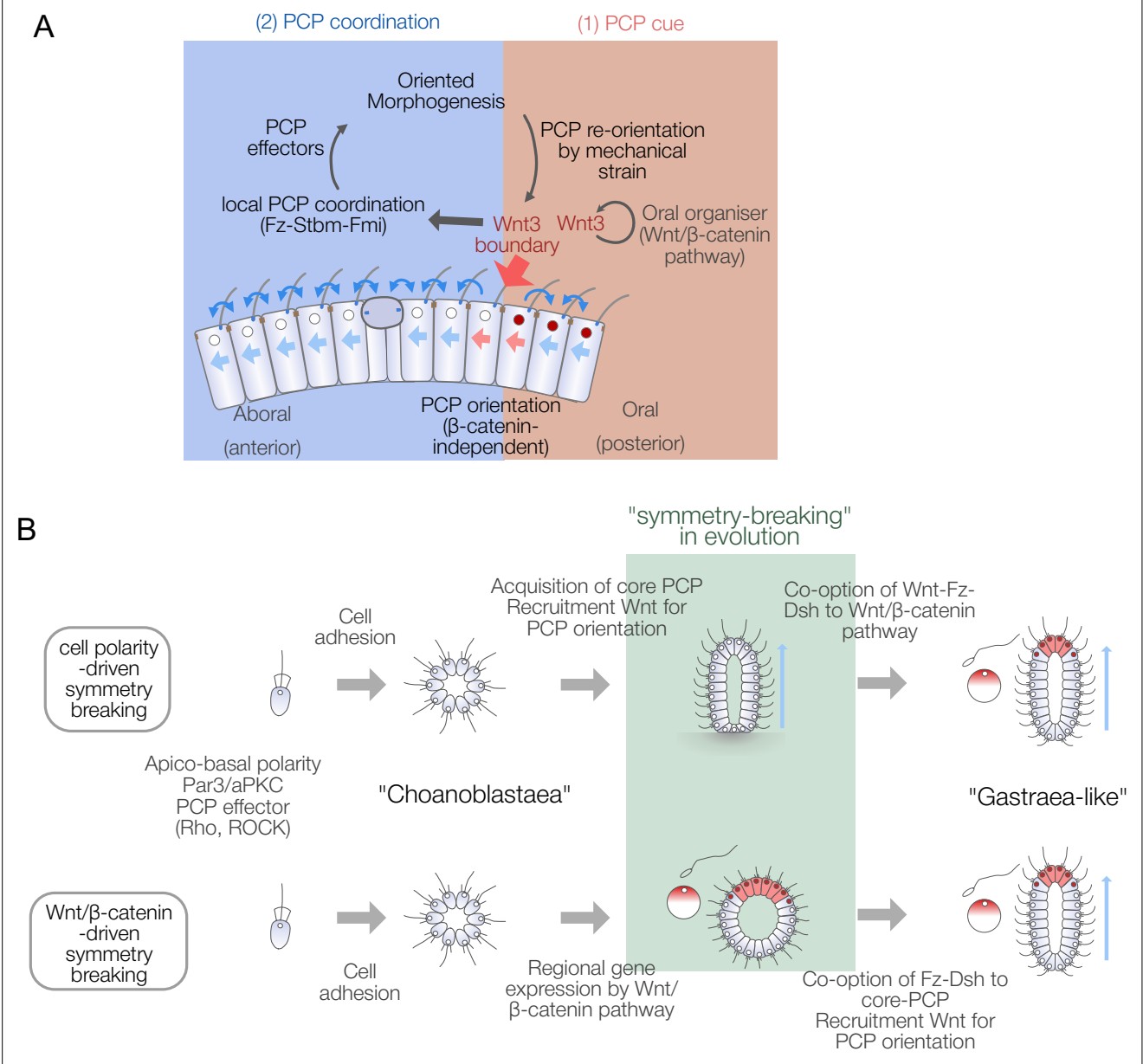

**Figure 6.** Models for body axis symmetry-breaking during *Clytia* embryogenesis and planar cell polarity (PCP)-driven axis innovation scenarios in metazoan evolution.

The online version of this article includes the following figure supplement(s) for figure 6:

**Figure supplement 1.** Fat-like and Dachsous play roles in long-range planar cell polarity (PCP) coordination for axial patterning in *Clytia* embryos.

**Figure supplement 2.** A wide heterochronic variation of axial elongation in phylum Cnidaria.

**Figure supplement 3.** Planar cell polarity (PCP), or coordinated polarity of motile cilia, is tightly coupled to axial patterning across metazoans and has independently evolved in non-metazoan species.

investigate the mechanism by which core PCP modulates axial gene expression. In *Clytia*, PCP defects resulting from Stbm knockdown led to changes in gene expression at the EG stage, notably affecting cells in the oral pole that ingress during gastrulation (*Lapébie et al., 2014*). Our work also connects to historical hypotheses regarding hydrozoan embryonic axis patterning mechanisms. Nearly a hundred years ago, Georges Teissier proposed that egg polarity is either maintained or re-established as a non-localised polarity within the embryo, thereby defining the body axis (*Teissier, 1931*). Later, Gary Freeman revisited Teissier's experiments and suggested that this polarity, somewhat consistent with

the modern concept of PCP, is directly inherited from egg polarity (*Freeman, 1981*; *Primus and Freeman, 2004*). Freeman considered this an alternative hypothesis to the maternal axis determinant model, which was later demonstrated in *Clytia* and other hydrozoans (*Momose et al., 2008*; *Momose and Houliston, 2007*). Our work highlights PCP propagation as a mechanistic explanation of Teissier and Freeman's non-localised polarity at gastrula and planula stages, while the two-step axis patterning model provides a link to localised Wnt/β-catenin pathway activation triggered by maternal determinants.

## Conservation of the PCP orientation and morphogenesis mechanisms in metazoans

In the *Drosophila* wing disc, local membrane-tethered Wg/Wnt4 in the wing margin can globally orient the PCP (*Wu et al., 2013*; *Yu et al., 2020*). In vertebrate tissues, Wnt ligands either act as global cues for PCP orientation (*Chu and Sokol, 2016*; *Gros et al., 2009*; *Minegishi et al., 2017*) or triggers involved in body elongation (*Heisenberg et al., 2000*; *Kilian et al., 2003*; *Tada and Smith, 2000*; *Wallingford and Harland, 2001*). The role of Wnt for PCP orientation is thus common across cnidarians and bilaterians. The Wnt subtypes employed in PCP orientation are, however, not conserved. The Wnt gene family gained its complexity prior to the bilaterian-cnidarian bifurcation (*Kusserow et al., 2005*). Then, different Wnt subtypes were likely co-opted in different taxa and developmental contexts. In zebrafish, PCP orientation by Wnt11 and core PCP regulation are distinctive in mechanosensory hair cell orientation (*Navajas Acedo et al., 2019*). The modulable two-step mechanism of Wnt-driven PCP orientation is likely common in metazoans.

Other cues than Wnt also function as PCP orientation cues in the animal kingdom. In the developing *Drosophila* eye and abdomen, another signalling module comprising the atypical cadherins Fat (Ft) and Dachsous (Ds) and kinase Four-jointed (Fj) regulates the localisation of core PCP complexes via gradients of Ds and Fj (*Strutt and Strutt, 2021*; *Yang et al., 2002*). Cnidarians have conserved Ft and Ds orthologue genes. In Hydra, fat-like protein (HyFat1) is implicated in epithelial cell alignment and adhesion (*Brooun et al., 2020*). In *Clytia*, inhibition of Fat-like (CheFat1) and Ds (CheDs) disrupted global PCP coordination, and the Ds disruption was restored by Wnt3 expression (*Figure 6—figure supplement 1*), suggesting that these molecules participate in global PCP orientation in a manner yet to be studied. Besides molecular signalling cues, mechanical constraints or fluid flow are also known to direct PCP (*Aigouy et al., 2010*; *Chien et al., 2022*; *Guirao et al., 2010*; *Mitchell et al., 2007*; *Olguín et al., 2011*) and can potentially act as the PCP symmetry-breaking factor. The availability of alternative PCP orientation cues might have allowed evolutionary flexibility to link PCP to available structures, such as the cell polarity scaffold of eggs. While tissue polarity orientation by PCP is strongly conserved, the cellular mechanisms underlying the morphogenetic movement are highly variable. Oriented cell division or convergent extension are likely major driving forces of blastula elongation prior to gastrulation in *Podocoryne carnea* (*Momose and Schmid, 2006*), another hydrozoan jellyfish. In contrast, in *Clytia*, elongation likely involves PCP-dependent intercalation of both ectoderm (*Byrum, 2001*) and ingressing endoderm (*van der Sande et al., 2020*) cells during gastrulation. In *Nematostella*, hydraulic pressure created by body wall muscle, organised along the body axis, drives the elongation (*Stokkermans et al., 2022*; *Figure 6—figure supplement 2*).

## Body axis symmetry-breaking in the evolutionary history of metazoa

The dual role of Wnt3 and the two-step PCP orientation mechanism revealed here are particularly intriguing in the context of understanding the emergence of animal body axes during evolution. The metazoan ancestor is considered to have taken a multicellular state such as a 'choanoblastaea' from a unicellular ancestor (*Arendt et al., 2015*; *Nielsen, 2008*; *Figure 6B*). Then, metazoans acquired a body axis and germ layers and formed a hypothetical ancestral state such as Haeckel's 'gastraea' (*Haeckel, 1873*) or Hyman's 'planuloid-acoeloid' (*Hyman, 1951*). Early multicellular ancestors thus acquired morphological axis patterning and coupled it with regional gene expression. It is tempting to speculate that the β-catenin/TCF-dependent local gene expression and morphogenetic patterning coordinated by core PCP proteins were independently acquired and coupled by co-opting Wnt ligands and signal transducers from one pathway to the other. For example, Wnt, Fz, and Dsh might have first been employed in the Wnt/β-catenin pathway, then recruited to a primordial PCP regulatory system (*Figure 6B*). Genome analyses indicate that PCP effectors and a part of core PCP components such

**Table 1.** Presence and absence of Wnt/β-catenin and core planar cell polarity (PCP) protein genes in basal metazoan and non-metazoan species based on existing studies.

Homoscleromorpha sponges (*Oscarella* spp.) possess a complete set of core PCP proteins. PCP regulated by core PCP proteins may have originated in a sponge-cnidarian-bilaterian common ancestor, which is equivalent to the metazoan common ancestor in the sponge-ancestor scenario. The absence of PCP effectors in fungi and *Capsaspora*, and the lack of *Inversin* and *Prickle* orthologues in choanoflagellates, suggests that primitive PCP regulatory mechanisms predate the metazoan ancestor. Symbols: +: orthologue identified with fully conserved domains; −: orthologue absent; ±: gene with homologous sequence present but lacking complete domain structure.

| | | | Wnt/β-catenin | | | | | Core PCP | | | | | | Ft/Ds/Fj | | |
| | | β-Catenin | Axin | GSK3-β | TCF | Wnt | Frizzled/Fzd | Dishevelled/Dvl | Flamingo/Celsr | Strabismus/Vangl | Inversin | Prickle | Fat/Fat-like | Dachsous/Ds | Rho/ROCK |
|---|---|---|---|---|---|---|---|---|---|---|---|---|---|---|---|
| Amoeba | *Dictyostelium discoideum* | ±* | | + | | | | | | | | | | | |
| Fungi | *Allomyces macrogynus* | ±§ | | + | | | | | | | | | | | + |
| Filasterea | *Capsaspora owczarzaki* | ±§ | | + | | | | | | | | | | | |
| Choanoflagellate | *Salpingoeca rosetta* | ±§ | | + | ±† | | | | − | | + | + | | | + |
| | *Monosiga brevicollis* | | | + | ±‡ | − | − | | − | − | + | +¶ | ±** | | + |
| Ctenophora | *Mnemiopsis leidyi* | + | − | + | + | + | + | | − | − | − | +¶ | | | + |
| Homoscleromorpha | *Oscarella carmela* | + | + | + | + | + | + | + | + | + | + | +¶ | ±** | | + |
| | *Oscarella lobularis* | + | | + | + | + | + | + | + | + | + | +¶ | | | + |
| Calcarea | *Sycon ciliatum* | + | − | + | + | + | + | + | + | − | + | +¶ | | | + |
| Demospongiae | *Amphimedon queenslandica* | + | + | + | + | + | + | + | − | − | + | +¶ | + | | + |
| Hexactinellida | *Aphrocallistes vastus* | + | + | + | + | − | + | − | − | − | + | +¶ | | | + |
| Anthozoa | *Nematostella vectensis* | + | + | + | + | + | + | + | + | + | + | +¶ | + | + | + |
| | *Clytia hemisphaerica* | + | + | + | + | + | + | + | + | + | + | + | +†† | + | + |
| Hydrozoa | *Hydra vulgaris* | + | + | + | + | + | + | + | + | + | + | + | +†† | + | + |
| Placozoa | *Tricoplax adherans* | + | + | + | + | + | + | + | + | + | − | + | + | + | + |
| Bilaterians | | + | + | + | + | + | + | + | + | + | + | + | + | + | + |

*β-Catenin in *Dictyostelium* is a β-catenin-like protein called Aardvark, a component of the junctional complex involved in cell signalling (not mediated by GSK3).

†TCF-like gene in choanoflagellates contains an HMG domain but is not a true TCF orthologue.

‡Dsh-like gene in choanoflagellates is a PDZ domain-containing protein lacking DEP and DIX domains.

§Choanoflagellate β-catenin is an armadillo repeat-containing protein.

¶*Prickle* genes in ctenophores and sponges derive from a common *Prickle-Testin* ancestor. Choanoflagellates possess *Testin* orthologues but not *Prickle*.

**Monosiga* and *Amphimedon* have Lefftyrin cadherins that share EGF and LamG domains with *Fat* cadherins, but their domain arrangement is different – the EGF/LamG domains are adjacent to the transmembrane domain rather than at the N-terminus.

††*Clytia* and *Hydra* have *Fat1* (Fat-like) genes but not *Fat4*. Refer to **Supplementary file 1** for the list of publications and gene accession numbers.

as Cdc42, RhoA, ROCK, Prickle, or Inversin are evolutionarily older than metazoans (*Lapébie et al., 2011*; *Table 1*, *Supplementary file 1*). Wnt-Fz-Dsh would then have been recruited to create morphogenesis machinery directed by PCP based on the existing local gene expression by Wnt/β-catenin signalling. Alternatively, core PCP interactions and Wnt-driven PCP orientation may be more ancestral with Wnt-Fz-Dsh used in PCP orientation prior to its involvement in 'canonical' Wnt signalling, acquired later through recruitment of the cell adhesion molecule β-catenin and its interaction with the DNA-binding TCF. Comparisons of the genomic repertoires for these components between species from across the metazoan phylogeny do not allow us to conclude which modules were acquired first (*Figure 6—figure supplement 3*, *Table 1*, *Supplementary file 1*). In the outgroups of the cnidaria-bilateria clade, a full set of core PCP proteins are present in a sponge *Oscarella carmela* (*Lapébie et al., 2011*; *Nichols et al., 2006*; *Schenkelaars et al., 2016*), and full Wnt/β-catenin pathways were identified in both Porifera and Ctenophora (*Holzem et al., 2024*), while multiple cases of loss of Stbm/Fmi occurred in Porifera and potentially in Ctenophora (*Figure 6—figure supplement 3*).

PCP is a fundamental character in multicellular tissue organisation seen wide across metazoans and may have been particularly important in the metazoan ancestor with lower anatomical complexity. It is known that the rotational polarity of the flagellar basal body is coordinated across the spheroid of green alga *Volvox carteri*, which thus independently acquired PCP-like character (*Hoops, 1993*; *Figure 6—figure supplement 3*), suggesting that a PCP equivalent evolved independently in *Volvox*. PCP-driven axis evolution is an attractive scenario and can explain how an initial 'choanoblastaea' multicellular ancestor acquired coordinated cilia/flagella flow. For example, PCP regulation might initially have improved filter-feeding efficiency in a choanoblastaea ancestor (*Cavalier-Smith, 2017*; *Nielsen, 2008*). Little notice has been paid to the implication of the PCP pathway and cellular basis of tissue patterning in metazoan body plan evolution compared to the Wnt/β-catenin pathway. A comparative cell biology, or 'Cellular Evo-Devo' approach, of the PCP/tissue polarities in diverse metazoans, in particular 'basal metazoans', including Cnidaria, Ctenophora, and Porifera, will be the key to addressing how symmetry-breaking to create a body axis arose during evolution.

# Materials and methods

## Key resources table

| Reagent type (species) or resource | Designation | Source or reference | Identifiers | Additional information |
|---|---|---|---|---|
| Sequence-based reagent | Wnt3MO | *Momose et al., 2008* | Morpholino oligonucleotide | CCAAAACACACCAGTGTCGAGCCAT |
| Sequence-based reagent | Fmi-MO | This paper | Morpholino oligonucleotide | CCTCAAGCCATCTGAGCTTCATTTT |
| Sequence-based reagent | Stbm-MO | *Momose et al., 2012* | Morpholino oligonucleotide | TCACTCCATCATCAAAATCATCCAT |
| Sequence-based reagent | Fat-like-MO | This paper | Morpholino oligonucleotide | CATCAAAATGTGAGACTTACCAGCC |
| Sequence-based reagent | Ds-MO | This paper | Morpholino oligonucleotide | TACGGTGATGGATAGTTCATCTTTC |
| Chemical compound, drug | Phalloidin Alexa Fluor 488 | Invitrogen | A12379 | 4 units/ml 0.13 µM in PBS |
| Chemical compound, drug | Phalloidin Alexa Fluor 647 | Invitrogen | A22287 | 4 units/ml 0.13 µM in PBS |
| Antibody | anti-Par3 (Rabbit polyclonal) | Merk Millipore | Cat# 07–330 RRID:AB_2101325 | IF (1:200) |
| Antibody | anti-γ Tubulin (Mouse monoclonal) | SIGMA | Cat# T5326 RRID:AB_532292 (GTU-88) | IF (1:500) |
| Antibody | anti-centrin (Mouse monoclonal) | Merck Millipore | Cat# 04–1624 RRID:AB_10563501 (20H5) | IF (1:200) |
| Recombinant DNA reagent | pCX3-Wnt3 | *Momose et al., 2008* | XLOC_001931 | Wildtype (3 bp mismatch in MO target) |

*Continued on next page*

*Continued*

| Reagent type (species) or resource | Designation | Source or reference | Identifiers | Additional information |
|---|---|---|---|---|
| Recombinant DNA reagent | pCX3-dnTCF | This paper | XLOC_007658 | Dominant negative N-terminal 27 a.a. deletion |
| Recombinant DNA reagent | pCX3-CA-β-cat | This paper | XLOC_003560 | Constitutively active S97A, T101A, S105A |
| Recombinant DNA reagent | pCX3-RhoABC-DN | This paper | XLOC_044216 | Dominant negative T19N |
| Recombinant DNA reagent | pCX3-RhoABC-CA1 | This paper | XLOC_044216 | Constitutively active G14V |
| Recombinant DNA reagent | pCX3-RhoABC-CA2 | This paper | XLOC_044216 | Constitutively active Q63L |
| Recombinant DNA reagent | pMiniTol2-ACT2::PH-Venus | This paper | U09117.1 | PLC-delta-1 PH-domain-Venus fusion protein |
| Recombinant DNA reagent | pMiniTol2-ACT2::Poc1-CC | This paper | HM010924.1 | *Clytia* Poc1-mCherry fusion protein |
| Strain, strain background | *Clytia hemisphaerica* Z4C[2] | EMBRC-Fr | Z4C2 | Male wildtype strain, discontinued |
| Strain, strain background | *Clytia hemisphaerica* Z4B | EMBRC-Fr | Z4B | Female wildtype strain, discontinued |
| Strain, strain background | *Clytia hemisphaerica* Z23 | EMBRC-Fr | Z23 | Male wildtype strain: (Z4C[2] × Z4B) |
| Strain, strain background | *Clytia hemisphaerica* Z30 | EMBRC-Fr | Z30 | Female wildtype strain: (Z4C[2] × Z4B) |
| Software, algorithm | ImageJ | ImageJ | RRID:SCR_003070 | |
| Software, algorithm | MATLAB | Mathworks | RRID:SCR_001622 | |
| Software, algorithm | Imaris | Oxford Instruments | RRID:SCR_007370 | |

## Animal culture and embryo preparation

Animal culture methods are described in *Lechable et al., 2020*. *C. hemisphaerica* laboratory strains were provided by the Service Aquariologie of CRB (IMEV-FR3761, Sorbonne Université/CNRS contact: Axel Duchene axel.duchene@imev-mer.fr) as an EMBRC-Fr service (https://www.embrc-france.fr/). Daily spawning was induced by light with a 24 hr day-night cycle. Eggs and sperm were separately collected from jellyfish and transferred into 10 cm glass dishes 2 hr after the light stimulation. Eggs were fertilised less than 1 hr after the spawning. To acquire EG stages, embryos were incubated at 16°C for 18 hr, which corresponds to the 12–13 hpf stage at the standard culture temperature (18°C).

## Microinjection

Morpholinos and mRNA were microinjected into unfertilised eggs or blastomeres at the 4-cell to 16-cell stages using an Eppendorf Femtojet microinjector with microcapillaries pulled with a Narishige PN-30 needle puller. The tip of the needle was broken by touching it to a glass surface scratched by a diamond pen to open the needle tip to give an outer diameter of 2–4 µm. Eggs were aligned in a microinjection chamber made by casting and polymerising PDMS in a negative mould printed with a Form3 3D printer. 2–3% of the eggs/cell volume of solution was injected by inserting a microcapillary needle with constant back-pressure (Femtojet compensation pressure). It typically takes 0.5–1 s for each egg at 5–10 kPa when the needle has an appropriate opening size. The volume of injection for each injection was visually controlled under an Olympus SZX6 stereomicroscope with SZ-ILLT oblique illumination. 3–5 mg/ml lysine fixable 10 kDa Dextran labelled with Alexa Fluor 647 or Cascade Blue

was co-injected with Wnt3 mRNA and Stbm-MO/Fmi-MO for the lineage tracking in the Wnt3-rescue experiment and PCP mosaic experiments.

## Plasmid construction

Mutant and fluorescent protein templates were constructed in pCX3 vector, which contains short 5' and 3' UTR sequences from CheStbm mRNA. Wildtype CheWnt3 with three mispaired nucleotide residues at the Morpholino target and CheTCF lacking amino acid residues 2–27 (dnTCF) as a dominant negative form (lacking β-catenin binding site) of human TCF-4 (*Korinek et al., 1997*) were amplified by PCR and cloned into the vector. A constitutively active form of Che-β-catenin (CA-β-cat) was created by introducing S97A, T101A, and S105A mutations corresponding to the constitutively active mice β-catenin mutant lacking a GSK3-β phosphorylation target (*Baba et al., 2006*). Constitutively active forms (CA1: G14V, CA2: Q63L) that gave identical phenotypes and a dominant form (DN: T19N) of CheRhoABC were designed based on the identical vertebrate RhoA mutants with the known phenotypes. These point mutations were introduced into wildtype cDNA clones in pCX3 vector using the QuickChange site-directed mutagenesis kit (Stratagene). Poc1-mCherry and PH-domain-Venus fusions were constructed in a transgenic pMiniTol2-ACT2 vector (*Weissbourd et al., 2021*) using NEBuilder Assembly kit from individually amplified PCR products or synthetic genes. mRNA was synthesised in vitro using mMessage mMachine T3 kit and poly(A)-tailing kit (Thermo Fisher) from PCR products amplified from the plasmids.

## Immunostaining

Immunofluorescent visualisation of PCP by anti-γ-tubulin antibody and phalloidin staining was described previously (*Momose et al., 2012*). The list of antibodies is in the Key resources table. In brief, the embryos were fixed with 4% paraformaldehyde in 0.1 M HEPES, 50 mM EGTA, 80 mM maltose, 10 mM MgSO$_4$, 0.1% Triton X-100, and washed in PBST (1x PBS with 0.1% Triton X-100), then incubated in antibody solution in 1% BSA fraction V in PBST, followed by three times of washes for at least 5 min in PBST. After removing the detergent by washing in PBS, actin was stained with fluorescently labelled phalloidin in PBS (0.13 μM). After staining, embryos were cleared and mounted in Citiflour AF-1 diluted with an equal volume of PBS and scanned using Leica SP5 or Stellaris confocal microscopes with ×63 objectives. The surface of the ciliated epidermis was Z-scanned (up to 10 μm) in the entire 246×246 μm$^2$ field of view and z-stacked to visualise cell contour with apical cortical actin and basal body, typically at 0.081 μm resolution in XY-axis and 0.5–1 μm resolution in Z-axis.

## Image analysis

The x-y coordinates of the apical surface centroid ($x_c$, $y_c$) of each cell and basal body of its cilium ($x_b$, $y_b$) are identified from the z-stacked confocal images. Measurement data, image treatment protocols, and ImageJ script are available from the Open Science Framework (https://doi.org/10.17605/OSF.IO/5PBGH). The polarity angle θ for each cell was calculated as

$$\theta = atan2\left(y_c - y_b, x_c - x_b\right)$$

A cell with its basal body and apical centroid less than 1.0 μm was treated as a non-polarised cell and excluded from the orientation analysis. The mean orientation (arrow in the radar plot) and circular standard deviation (circ. s.d.) are calculated (*Berens, 2009*). A MATLAB script was used to summarise the PCP orientation and statistics (see the OSF repository). The morphology of embryos in three dimensions (*Figure 5D and H*) is reconstructed from a series of cortical actin confocal images by building a surface of the volume containing the cortical actin signals using Imaris software (Oxford Instruments).

## Acknowledgements

This work was supported by Agence Nationale de la Recherche (ANR-09-BLAN-0236-01 and ANR-17-CE13-0016-01), CNRS INSB Diversity of Biological Mechanisms (DBM) programme and Sorbonne Université SATTSU André Picard Network. We thank the Aquariology Service of CRB (IMEV – FR 3761) for access to marine facilities, biological resources, and technical advice. This service is supported by EMBRC-France, whose French state funds are managed by the ANR within the Investments of the

Future programme under reference ANR-10-INBS-02. We also thank IMEV PIM for access to microscopy equipment. We thank Brandon Weissbourd for the critical reading of the manuscript. We thank Fumiaki Sugahara and Shigeru Kuratani for their suggestions and comments on evolutionary theories.

## Additional information

### Funding

| Funder | Grant reference number | Author |
| --- | --- | --- |
| Agence Nationale de la Recherche | ANR-09-BLAN-0236-01 | Evelyn Houliston |
| Agence Nationale de la Recherche | ANR-17-CE13-0016-01 | Tsuyoshi Momose |
| Centre National de la Recherche Scientifique | Diversity of Biological Mechanisms | Tsuyoshi Momose |
| Sorbonne Université | SATTSU Réseau André Picard | Christine Vesque |

The funders had no role in study design, data collection and interpretation, or the decision to submit the work for publication.

### Author contributions

Julie Uveira, Marion Lechable, Data curation, Investigation, Methodology; Antoine Donati, Marvin Léria, Data curation, Investigation; François Lahaye, Methodology; Christine Vesque, Funding acquisition; Evelyn Houliston, Funding acquisition, Writing - review and editing; Tsuyoshi Momose, Conceptualization, Data curation, Supervision, Funding acquisition, Investigation, Methodology, Writing – original draft, Project administration

### Author ORCIDs

Antoine Donati  http://orcid.org/0000-0002-3616-3915
Marvin Léria  http://orcid.org/0009-0006-5287-4123
Marion Lechable  https://orcid.org/0009-0000-5307-7608
Christine Vesque  https://orcid.org/0000-0001-7983-4953
Evelyn Houliston  https://orcid.org/0000-0001-9264-2585
Tsuyoshi Momose  https://orcid.org/0000-0002-3806-3408

Reviewer #1 (Public review): https://doi.org/10.7554/eLife.104508.3.sa1
Reviewer #2 (Public review): https://doi.org/10.7554/eLife.104508.3.sa2
Author response https://doi.org/10.7554/eLife.104508.3.sa3

## Additional files

### Supplementary files

MDAR checklist

Supplementary file 1. Information on the presence of genes in key species, including accession numbers from sequence databases, or citations of studies reporting their absence.

### Data availability

Raw data of the cell polarity measurements are available from https://osf.io/5pbgh/.

The following dataset was generated:

| Author(s) | Year | Dataset title | Dataset URL | Database and Identifier |
| --- | --- | --- | --- | --- |
| Momose T | 2025 | Clytia_Planar-Cell-Polarity_measurmrent | https://osf.io/5pbgh/ | Open Science Framework, 5pbgh |

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
