## [Editor Report · eLife Assessment]

This analysis of the formation of the oral-aboral body axis in cnidarians, the sister group of bilaterians, is a significant and **fundamental** contribution to the field of Wnt signalling and planar cell polarity, particularly in or understanding in gradient formation, non-canonical Wnt signalling and Wnt-Frizzled interactions in cnidarians. The evidence supporting the conclusions is **compelling** and has the potential to contribute to a deeper understanding of the origin and evolution of Wnt signalling in cnidarians and metazoans in general. These findings, which are presented in a thoughtful and scholarly manner, will be of broad interest to developmental and evolutionary biologists.

---

## [Referee Report · Reviewer #1 (Public review)]

Summary:

This noteworthy paper examines the role of planar cell polarity and Wnt signalling in body axis formation of the hydrozoan Clytia. In contrast to the freshwater polyp Hydra or the sea anemone Nematostella, Clytia represents a cnidarian model system with a complete life cycle (planula larva-polyp-medusa). In this species, classical experiments have demonstrated that a global polarity is established from the oral end of the embryos (Freeman, 1981). Prior research has demonstrated that Wnt3 plays a role in the formation of the oral organiser in Clytia and other cnidarians, acting in an autocatalytic feedback-loop with β-catenin. However, the question of whether and to what extent an oral-aboral gradient of Wnt activity is established remained unanswered. This gradient is thought to control both tissue differentiation and tissue polarity. The planar cell polarity (PCP) pathway has been linked to this polarity, although it is generally considered to be β-catenin independent.

Comments on major strengths and weaknesses:

Beautiful and solid experiments to clarify the role of canonical Wnt signalling and PCP core factors in coordinating planar cell polarity of Clytia. The authors have conducted a series of sophisticated experiments utilising morpholinos, mRNA microinjections and immunofluorescent visualisation of PCP. The objective of these experiments was to address the function of Wnt3, β-catenin and PCP core proteins in the coordination of the global polarity of Clytia embryos. The authors conclude that PCP plays a role in regulating polarity along the oral-aboral axis of embryos and larvae. This offers a conceivable explanation for how polarity information is established and distributed globally during Clytia embryogenesis, with implications for our understanding of axis formation in cnidarians and the evolution of Wnt signalling in general. - While the experiments are well-designed and executed, there are some criticisms, questions or suggestions that should be addressed.

(i) Wnt3 cue and global PCP. PCP has been described in detail in a previous paper on Clytia (Momose et al, 2012): its orientation along the oral-aboral body axis (ciliary basal body positioning studies), and its function in directional polarity during gastrulation (Stbm-, Fz1-, and Dsh-MO experiments). I wonder if this part could be shortened. What is new, however, are the knockdown and Wnt3-mRNA rescue experiments, which provide a deeper insight into the link between Wnt3 function in the blastopore organiser as a source or cue for axis formation. These experiments demonstrate that the Wnt3 knockdown induces defects equivalent to PCP factor knockdown, but can be rescued by Wnt3-mRNA injection, even at a distance of 200 µm away from the Wnt-positive area. The experimental set-up of these new molecular experiments follows in important aspects those of Freeman's experiments of 1981 (who in turn was motivated to re-examine Teissier's work of 1931/1933 ...). Freeman did not use the term "global polarity" but the concept of an axis-inducing source and a long-range tissue polarity can be traced back to both researchers.

(ii) PCP propagation and β-catenin. The central but unanswered question in this study focuses on the interaction between Wnt3 and PCP and the propagation of PCP. Wnt3 has been described in cnidarians but also in vertebrates and insects as a canonical Wnt interacting with β-catenin in an autocatalytic loop. The surprising result of this study is that the action of Wnt3 on PCP orientation is not inhibited in the presence of a dominant-negative form of CheTCF (dnTCF) ruling out a potential function of β-catenin in PCP. This was supported by studies with constitutively active β-catenin (CA-β-cat) mRNA which was unable to restore PCP coordination nor elongation of Wnt3-depleted embryos but did restore β-catenin-dependent gastrulation. Based on these data, the authors conclude that Wnt3 has two independent roles: Wnt/β-catenin activation and initial PCP orientation (two step model for PCP formation). However, the molecular basis for the interaction of Wnt3 with the PCP machinery and how the specificity of Wnt3 for both pathways is regulated at the level of Wnt-receiving cells (Fz-Dsh) remains unresolved. - Also, with respect to PCP propagation, there is no answer with respect to the underlying mechanisms. The authors found that PCP components are expressed in the mid-blastula stage, but without any further indication of how the signal might be propagated, e.g., by a wavefront of local cell alignment. Here, it is necessary to address the underlying possible cellular interactions more explicitly.

(iii) The proposed two step model for PCP formation has important evolutionary implications in that it excludes the current alternate model according to which a long-range Wnt3-gradient orients PCP ("Wnt/β-catenin-first"). Nevertheless, the initial PCP orientation by Wnt3 - as proposed in the two-step-model - is not explained at all on the molecular level. Another possible, but less well discussed and studied option for linking Wnt3 with PCP action could be a role of other Wnt pathways. The authors present compelling evidence that Wnt3 is the most highly expressed Wnt in Clytia at all stages of development. The authors convincingly show that Wnt3 is the most highly expressed Wnt in Clytia at all stages of development (Fig. S1). However, Wnt7 is also more highly expressed, which makes it a candidate for signal transduction from canonical Wnts to PCP Wnts. An involvement of Wnt7 in PCP regulation has been described in vertebrates (http://dx.doi.org/10.1016/j.celrep.2013.12.026). This would challenge the entire discussion and speculation on the evolutionary implications according to which PCP Wnt signaling comes first (PCP-first scenario") and canonical Wnt signaling later in metazoan evolution.

(iv) The discussion, including Figure 6, is strongly biased towards the traditional evolutionary scenario postulating a choanzoan-sponge ancestry of metazoans. Chromosome-linkage data of pre-metazoans and metazoans (Schulz et al., 2023; https://doi.org/10.1038/s41586-023-05936-6) now indicate a radically different scenario according to which ctenophores represent the ancestral form and are sister to sponges, cnidarians and bilaterians (the Ctenophora-sister hypothesis). This also has implications for the evolution of Wnt signalling, as discussed in the recent Nature Genetics Review by Holzem et al. (2024) (https://doi.org/10.1038/s41576-024-00699-w). Furthermore, it calls into question the hypothesis of a filter-feeding multicellular gastrula-like ancestor as proposed by Haeckel (Maegele et al., 2023). These papers have not yet been referenced, but they would provide a more robust discussion.

General appraisal:

The authors have carefully addressed all important points raised in this review. Aims and results support their conclusions.

Impact of the work, utility of methods and data:

As stated above, there will be a major impact on our understanding of the role of Wnt signaling in gradient formation and particularly the role of non canonical wnt signaling. As mentioned above, this will have a major impact on our understanding of the role of Wnt signalling in gradient formation, particularly the role of non-canonical Wnt signalling. - It will also be important to better understand the role of Wnt-Frizzled interactions in these basal organisms, as cnidarians have a smaller repertoire of Frizzled receptors compared to the relatively complete repertoire of Wnt subfamilies. This may imply that Wnt 3 is active in both canonical and PCP.

Additional context:

With regard to the question of the evolution of the body plan and Wnt signalling, it would be helpful and important for readers unfamiliar with cnidarians to know that the Hydrozoa/Medusozoa, to which Clytia belongs, are an "evolutionary derived group" within the Cnidaria, as opposed to the Anthozoa (e.g. sea anemone Nematostella). Hydrozoans possess planula larvae that are devoid of a mouth and any form of feeding mechanism, relying instead on the yolk of a fertilised egg for sustenance. The substantial divergence between the Anthozoa and Medusozoa was accompanied by significant gene reductions within the Medusozoa, which likely exerts an influence on the evolution of Wnt signalling in this group as well. This should not detract from the value of the work, but may help to put it in perspective.

---

## [Referee Report · Reviewer #2 (Public review)]

Summary:

Canonical Wnt signaling has previously been shown to be responsible for correct patterning of the oral-aboral axis as well as germ layer formation in several cnidarians. The post-gastrula stage, the planula larvae is not only elongated, it has a specific swimming direction due to the decentralized cellular positioning and slanted anchoring of the cilia. This, in turn, is in most other animals the result of a Wnt-Planar-cell polarity pathway. This paper by Uveira et al investigates the role of Wnt3 signaling in serving as a local cue for the PCP pathway which then is responsible for the orientation of the cilia and elongation of the planula larva of the hydrozoan Clytia hemisphaerica. Wnt3 was shown before to activate the canonical pathway via ß-catenin and to act as an axial organizer. The authors provide compelling evidence for this somewhat unusual direct link between the pathways through the same signaling molecule, Wnt3. In conclusion, they propose a two-step model: (1) local orientation by Wnt3 secretion (2) global propagation by the PCP pathway over the whole embryo.

Strengths:

In a series of elegant and also seemingly sophisticated experiments, they show that Wnt3 activates the PCP pathway directly, as it happens in the absence of canonical Wnt signaling (e.g. through co-expression of dnTCF). Conversely, constitutive active ß-catenin was not able to rescue PCP coordination upon Wnt3 depletion, yet restored gastrulation. This uncouples the effect of Wnt3 on axis specification and morphogenetic movements from the elongation via PCP. Through transplantation of single blastomeres providing a local source of Wnt3, they also demonstrate the reorganization of cellular polarity immediately adjacent to the Wnt3 expressing cell patch. These transplantation experiments also uncover that mechanical cues can also trigger the polarization, suggesting a mechanotransduction or direct influence on subcellular structures, e.g. actin fiber orientation.

This is a beautiful and elegant study addressing an important question. The results have significant implications also for our understanding of the evolutionary origin of axis formation and the link of these two ancient pathways, which in most animals are controlled by distinct Wnt ligands and Frizzled receptors. The quality of the data is stunning and the paper is written in a clear and succinct manner. This paper has the potential to become a widely cited milestone paper.

Weaknesses:

I can not detect any major weaknesses. The work only raises a few more follow-up questions, which the authors are invited to comment on.

I acknowledge the revisions made by the authors. Some open questions remain that need to be addressed in future work, and I accept the limitations of this study, as argued by the authors. Besides the elegant and high-quality experiments, I also appreciate the thoughtful and inspiring discussion.

---

## [Author Response]

The following is the authors’ response to the original reviews

**Reviewer #1 (Public review):**
(1) Wnt3 cue and global PCP. PCP has been described in detail in a previous paper on Clytia (Momose et al, 2012): its orientation along the oral-aboral body axis (ciliary basal body positioning studies), and its function in directional polarity during gastrulation (Stbm-, Fz1-, and Dsh-MO experiments). I wonder if this part could be shortened. What is new, however, are the knockdown and Wnt3-mRNA rescue experiments, which provide a deeper insight into the link between Wnt3 function in the blastopore organiser as a source or cue for axis formation. These experiments demonstrate that the Wnt3 knockdown induces defects equivalent to PCP factor knockdown, but can be rescued by Wnt3-mRNA injection, even at a distance of 200 µm away from the Wnt-positive area. The experimental set-up of these new molecular experiments follows in important aspects those of Freeman's experiments of 1981 (who in turn was motivated to re-examine Teissier's work of 1931/1933 ...). Freeman did not use the term "global polarity" but the concept of an axis-inducing source and a long-range tissue polarity can be traced back to both researchers.

We appreciate the reviewer’s insightful comments for evolutionary biology and cnidarian developmental biology.

Concerning the presentation of the basic PCP structure of *Clytia* embryo epidermal cells, we prefer to retain this section unless there is a strict limit on manuscript length. These experiments provide background information necessary to establish the biological system for the readers. The structures of cells, notably cell adhesion, cilia, and the cytoskeleton, are essential components of this system.

We have restored sentences concerning the historical contributions of Freeman and Teissier from a previous version of the manuscript.

Freeman’s work offered two key insights. The first is the concept that cell polarity spreads and self-organizes over the distances revealed by the tissue orientation of aggregate embryonic cells (Freeman, 1981 https://doi.org/10.1007/BF00867804), which was termed “global polarity” in a review by Primus and Freeman (2004 https://doi.org/10.1002/bies.20031). This concept closely resembles the modern understanding of PCP coordination mechanisms mediated by core PCP interactions. Remarkably, Freeman proposed this idea in the early 1980s, at the same time of the first characterization of PCP mutants in *Drosophila* (Gubb and Garcia-Bellido 1982). The second is the role of egg polarity in defining the axis. Freeman demonstrated that the position of the first cleavage furrow predicts the oral-aboral axis by a series of sophisticated experiments. This was a milestone for the studies of cnidarian body axis development.

However, some of Freeman’s interpretations were misleading. In the 1981 paper, he stated:

"Polarity

Other work that I have done has established that the anterior-posterior axis of the planula is set up at the time of the first cleavage; the site where cleavage is initiated specifies the posterior pole of this axis (Freeman 1980). The experiment reported here in which embryos were cut into halves and each half regulated to form a normal planula with the same polarity properties as the embryo it is from provides evidence that these polarity properties are remarkably stable at all developmental stages tested ranging from 4 cell to postgastrula embryos. "

Freeman hypothesised that cell polarity at the 2- or 4-cell stage, referred to as the “polarity of first cell cleavage,” is directly inherited as the global polarity observed in later developmental stages.

In the review by Primus and Freeman (2004), two hypotheses were introduced: (1) maternally localised factors, such as mRNA, determine the axis, and (2) cell polarity of cleavage furrow formation, is inherited to later stages and determines the axis. Freeman described these two hypotheses as mutually exclusive. However, we now know that cell polarity at early cleavage stages does not directly contribute to global polarity/PCP. Instead, Wnt/β-catenin signaling is regionally activated by maternally localised mRNAs distributed along the egg polarity (Momose, 2007; Momose, 2008), which maintain Wnt3 localisation and direct morphological axis patterning. Our study shown in this article unified these hypotheses.

On the second point, as the reviewer noted, Freeman indeed revisited the work of Georges Teissier (Teissier, 1931), who conducted similar experiments on *Amphisbetia* embryos. It was Teissier who first described how the egg polarity is preserved in later stages and defines the axis. Teissier, however, carefully avoided asserting continuity between egg and blastula polarities, allowing for the possibility of “rétablissement” (re-establishment). As Teissier stated:

"…On constate, en second lieu, que la polarité de l’œuf se conserve dans chacun de se fragment et que le maintien ou le rétablissement de cette polarité sont indispensables à un développement normal. Un fragment d’œuf ou de morula n’a aucune partie ni aucun blastomère qui soit rigoureusement déterminé comme endoderme, mais possède, par contre, un pôle antérieur et un pôle postérieur bien définis.…

Mais cette proposition, qui ne semble pourtant guère dépasser la simple constatation des faits, soulève de grave difficulté. Elle donne en effet à la polarité, propriété encore bien mystérieuse, un rôle morphogénétique de premier ordre et implique des conséquences trop importantes pour qu’on puisse l’accepter sans un très sérieux examen.

Comme je ne pense pas que les questions relatives à la nature des localisation germinales, à l’existence et au fonctionnement des organisateurs de l’œuf des Cœlentérés, puissant, dans l’état actuel de nos connaissances, être discutées utilement, je ne veux voir dans la proposition précédente qu’une façons commode et tout provisoire de systématiser les faits."

English translation:

“We note also that the polarity of the egg is preserved in each fragment and that the maintenance or re-establishment of this polarity is essential for normal development. A fragment of egg or morula has no part or blastomere that is rigorously determined as endoderm, but has, on the other hand, a well-defined anterior and posterior pole....

But this proposition, which hardly seems to go beyond the simple observation of facts, raises serious difficulties. It gives polarity, still a mysterious property, a morphogenetic role of the first order, and implies consequences too important to be accepted without very serious examination.

As I do not believe that questions concerning the nature of germinal localisation, or the existence and functioning of the egg organisers in Coelenterates, can, in the present state of our knowledge, be usefully discussed, I prefer only to see in the foregoing proposition a convenient and very provisional way of systematising the facts.”

Teissier, G. (1931). Étude Expérimentale du Développement de Quelques Hydraires. Ann. Sc. Nat. Zool 14, 5–59.

Teissier's interpretation and caution were reasonable.

Our work connects recent molecular research on axis specification mechanisms in cnidarians with the classic experimental studies of Freeman and Teissier. We believe it is essential to present and acknowledge their conceptual contributions. We have updated the Discussion to include these points.

(2) PCP propagation and β-catenin. The central but unanswered question in this study focuses on the interaction between Wnt3 and PCP and the propagation of PCP. Wnt3 has been described in cnidarians but also in vertebrates and insects as a canonical Wnt interacting with β-catenin in an autocatalytic loop. The surprising result of this study is that the action of Wnt3 on PCP orientation is not inhibited in the presence of a dominant-negative form of CheTCF (dnTCF) ruling out a potential function of β-catenin in PCP. This was supported by studies with constitutively active β-catenin (CA-β-cat) mRNA which was unable to restore PCP coordination nor elongation of Wnt3-depleted embryos but did restore β-catenin-dependent gastrulation. Based on these data, the authors conclude that Wnt3 has two independent roles: Wnt/β-catenin activation and initial PCP orientation (two-step model for PCP formation). However, the molecular basis for the interaction of Wnt3 with the PCP machinery and how the specificity of Wnt3 for both pathways is regulated at the level of Wnt-receiving cells (Fz-Dsh) remain unresolved. Also, with respect to PCP propagation, there is no answer with respect to the underlying mechanisms. The authors found that PCP components are expressed in the mid-blastula stage, but without any further indication of how the signal might be propagated, e.g., by a wavefront of local cell alignment. Here, it is necessary to address the underlying possible cellular interactions more explicitly.

The question of how Wnt3 interacts with the core PCP complex remains open for future investigation. An obvious hypothesis is that one of the Frizzled receptors binds Wnt3 ligands. For additional details, please refer to the response to Reviewer 2’s comment. Regarding other non-classic Wnt receptors, studies in the developing mouse limb have demonstrated that a Wnt5a gradient controls PCP polarisation via ROR receptors and graded Strabismus phosphorylation (Gao et al., 2011, https://doi.org/10.1016/j.devcel.2011.01.001). However, in this context, the Wnt5a gradient influences the frequency of polarised cells rather than PCP orientation. In *Clytia*, we performed gene knockdown experiments targeting ROR and RYK receptors using Morpholinos but did not observe any effect on axial patterning, suggesting that these receptors are unlikely to be involved in Wnt3 interaction.

Concerning PCP propagation mechanisms, these are well-characterized in both *Drosophila* and vertebrates and conserved across taxa. The localised Fz-Fmi complex at the apical cortex of a cell interacts with the oppositely localised Stbm-Fmi complex in neighbouring cells, enabling coordination of PCP between directly adjacent cells. This interaction provides a comprehensive explanation for PCP propagation mechanisms. In Drosophila, the “domineering non-autonomy” effect is a well-documented phenomenon where PCP orientation autonomously propagates from core PCP mutant mosaic patches. Overall, PCP propagation is a conserved and robust mechanism across metazoans.

(3) The proposed two-step model for PCP formation has important evolutionary implications in that it excludes the current alternate model according to which a long-range Wnt3-gradient orients PCP ("Wnt/β-catenin-first"). Nevertheless, the initial PCP orientation by Wnt3 - as proposed in the two-step model - is not explained at all on the molecular level. Another possible, but less well-discussed and studied option for linking Wnt3 with PCP action could be the role of other Wnt pathways. The authors present compelling evidence that Wnt3 is the most highly expressed Wnt in Clytia at all stages of development. The authors convincingly show that Wnt3 is the most highly expressed Wnt in Clytia at all stages of development (Figure S1). However, Wnt7 is also more highly expressed, which makes it a candidate for signal transduction from canonical Wnts to PCP Wnts. An involvement of Wnt7 in PCP regulation has been described in vertebrates (http://dx.doi.org/10.1016/j.celrep.2013.12.026). This would challenge the entire discussion and speculation on the evolutionary implications according to which PCP Wnt signaling comes first (PCP-first scenario") and canonical Wnt signaling later in metazoan evolution.

First of all, we apologise that the expression profile of Wnt7originally provided in Figure S1 was incorrect; Wnt7 is not expressed in the embryonic stage. The error came from the accession number XLOC_034538 assigned to two transcripts, Wnt7 and Ataxin10, in the published genome assembly. Once the expression profile is revised in this light, the data are consistent with the in situ hybridisation data published in Momose et al. (2012, https://doi.org/10.1242/dev.084251). Wnt3 is the only Wnt ligand detectable between egg and gastrula stages. We appreciate the reviewer highlighting this issue and have corrected Figure S1

If we understand correctly, the reviewer raises the possibility that Wnt3's downstream canonical Wnt/β-catenin pathway activates the expression of other Wnt genes, which in turn orient the PCP. Indeed, we showed that the expression of Wnt1 (previously called WntX2), Wnt2 (WntX1A), Wnt5 and Wnt6 (Wnt9) all becomes undetectable at the planula stage following Wnt3-MO injection (Momose et al., 2012). So, it is a reasonable concern.

This possibility can be excluded because the canonical pathway activation by CA-β-cat does not restore PCP in Wnt3-MO-injected embryos and Wnt3 can orient PCP without Wnt/β-catenin pathway activity in the presence of dominant negative TCF (dnTCF). Concerning Wnt1b and Wnt11b, these transcripts are maternally stored and even more abundant than Wnt3. However, we can conclude that these do not have any role in axis patterning based on the complete axis loss in Wnt3-MO morphants.

Lastly, it should of course be remembered that the chronological order of characters appearing in a developmental process does not necessarily reflect their appearance in evolution from ancestral to modern.

(4) The discussion, including Figure 6, is strongly biased towards the traditional evolutionary scenario postulating a choanzoan-sponge ancestry of metazoans. Chromosome-linkage data of pre-metazoans and metazoans (Schulz et al., 2023; https://doi.org/10.1038/s41586-023-05936-6) now indicate a radically different scenario according to which ctenophores represent the ancestral form and are sister to sponges, cnidarians and bilaterians (the Ctenophora-sister hypothesis). This has also implications for the evolution of Wnt signalling, as discussed in the recent Nature Genetics Review by Holzem et al. (2024) (https://doi.org/10.1038/s41576-024-00699-w). Furthermore, it calls into question the hypothesis of a filter-feeding multicellular gastrula-like ancestor as proposed by Haeckel (Maegele et al., 2023). These papers have not yet been referenced, but they would provide a more robust discussion.

I overlooked the excellent work of Holzem and colleagues. I appreciate this suggestion. The work, unfortunately, focusses mainly on the Wnt/β-catenin pathway. The PCP pathway consists of not only core PCP (Fmi Stbm, Pk, Dgo, Fz and Dsh) but many other components, such as Rho GTPase, which are all dealt with as "PCP” in this review. While the full set of core PCP is present only in the phylum Cnidaria and bilaterians, Pk and Dgo are present in choanoflagellate and Rho GTPase or ROCK are present even in Fungi (Lapébie et al, 2011 DOI 10.1002/bies.201100023). Holzem et al., described PCP as absent in ctenophores, likely based on the lack of Fmi/Stbm, while claiming its presence in fungi based on Rho GTPase and ROCK. This led to their argument that the Wnt/β-catenin pathway is more ancestral, supported by the absence of PCP components in ctenophores alongside the ctenophore-sister hypothesis.

This likely reflects the limited attention given to PCP in the metazoan evolutionary biology community. Our work sheds light on the importance of PCP regulation in metazoan evolution. In the revised Discussion, we emphasise this point together with the importance of cell biology studies in basal metazoans and compare them based on functional studies.

The observation of *Aiptasia*’s predatory “gastrula-like” larvae is indeed fascinating. Understanding how early metazoan ancestors obtained nutrients is a key to uncovering the origins of metazoans. However, the relevance of this work to metazoan evolution remains unclear. Predatory nutrient uptake is common among cnidarians, and the findings of Maegele et al. could suggest that the predatory gastrula-like state is ancestral, with the symbiotic state being derived, within Cnidaria, but does not notably support it in metazoa. Also, it has to be clarified how predation is defined. Fundamentally, there is little distinction between filter-feeding and predatory feeding regarding heterotrophy; both feeding types require digestive machinery. If active feeding behaviour is the essence of predation, this would be better addressed as an evolutionary neurobiology or neuroscience question. Another mystery is what the metazoan ancestors took as food if they were predatory; there has to be a non-predatorial metazoan, as a food, already present before them.

Overall, Maegele’s work seems premature to be incorporated into the metazoan evo-devo discussion. In either case, the standard approach would involve comparative studies across taxa. It will be interesting to see follow-up works on comparative and functional genomics of predatory/digestive machinery within phylum cnidaria and across metazoan, including sponge and ctenophores.

**Reviewer #2 (Recommendations for the authors):**

We appreciate the reviewer’s expertise and recommendations regarding Wnt and PCP signalling. It would be our great pleasure if our work is seen and referenced by the cell biology community using model animals.

(1) According to the 2-step model, one would expect that there is a temporal gradient in the spreading of the PCP from oral to aboral. Is there any indication for this?

The best indication of a spatial and temporal gradient of PCP establishment observed so far is at the blastula stage (Fig.2B). PCP gradually becomes coordinated starting at 9 hpf, when PCP is slightly better organised close to the Wnt3-positive area (oral) compared to distal (aboral) areas. We did live imaging with tagged Poc1 to track the positions of centrioles in each cell (Fig. 2E), but this did not provide any further information about the spreading of the PCP. We hypothesise that there is a delay between PCP polarisation—established through the subcellular localisation of core PCP components—and its structural manifestation as ciliary positioning and orientation. This delay likely varies between cells, preventing the formation of a precise spatial PCP wave. We hope in the future to address this temporal aspect by live-imaging of core PCP proteins labelled with fluorescent proteins.

(2) PCP is likely to be an all-or-nothing effect, while axial patterning is dose-dependent. is there a critical dose of Wnt3 level required to kick off the PCP pathway?

We agree that the PCP phenotype is all-or-nothing. Although we did not perform a quantitative test, we have not seen any intermediate phenotypes in Wnt3-rescue experiments. In our experimental condition (100 ng/µl mRNA), the Wnt3 mRNA injection into a blastomere consistently restores the body axis (via PCP) of Wnt3-MO injected embryos. No axis restoration was observed at 1 ng/µl. At 10 ng/µl, some embryos showed a restored elongated axis, while others showed no axis. The volume of injection is not precisely controllable and can easily vary two-fold, so we assume the limit is somewhere around 10 ng/µl. This contrasts with endoderm rescue via Wnt/β-catenin activation by GSK-β-inhibitors (alsterpaullone) or the constitutively active β-catenin (CA-β-cat), which occurs in a dose-dependent manner (ex. Supplementary Figure S2).

(3) The key question left unaddressed is whether Wnt3 signals through one or two different Frizzled receptors? Which Frizzled receptors are candidates for this? Could they be knocked down to see which pathway (or both) is affected?

How Wnt3 orientates the PCP system is an extremely interesting question that needs to be answered, and we plan to address this in the future. In *Clytia*, four Frizzled genes have been identified in the genome: *CheFz1* (vertebrate counterpart of Fz1, 2, 3, 6 and 7), *CheFz2* (Fz5 and 8), *CheFz3* (Fz9/10) and *CheFz4* (Fz4). Knockdown of CheFz1, hereby called Fz1, by Morpholino showed a PCP phenotype (Momose 2012, supplementary data). For a long time, we have suspected that the most likely candidate for PCP mediation is CheFz1. The Wnt3-rescue experiment in CheFz1-blocked background (similar experiment to Figure 3E, F) could potentially have answered this question. No PCP orientation would be expected even near the Wnt3-mRNA injected area if CheFz1 was the Wnt3 receptor for PCP orientation. Unfortunately, no reliable PCP phenotype was observed in this experiment, so this experiment was not included in the manuscript. We initially thought this was due to incomplete suppression of CheFz1 mRNA translation by the Morpholino when used at sub-toxic doses. But we now favour the alternative explanation that Fz1 does not mediate the Wnt3 signal responsible for initiating PCP orientation. We have previously shown that Fz1 is required for the Wnt/ β-catenin pathway (indicated by nuclear β-catenin localisation Momose 2007), which is then required to maintain Wnt3 expression. We cannot rule out that the PCP phenotype obtained previously following Fz1 knockdown (supplementary data in Momose 2012) is an indirect effect of Wnt3 downregulation.

In future work, we plan to test the PCP involvement of the other *Clytia* Frizzleds, notably CheFz2 and CheFz4, which are not present as maternal mRNAs but are zygotically expressed in the early gastrula stage. CheFz3 is unlikely to be a candidate because it is aborally localised and acts as a negative receptor for the Wnt/β-catenin pathway (Momose 2007). Lastly, in unpublished experiments, no axial phenotype was obtained with ROR and RYK knockdown by Morpholino (T. Momose unpublished).

Based on these considerations, our current working hypothesis is that Wnt3 somehow stabilises or activates one of the Frizzled receptors acting as a core PCP protein in a polarised manner, likely at the oral side of each cell (Stbm is localised at the aboral side), which breaks the PCP symmetry and is propagated across the body axis.

A few lines have been added to the discussion regarding this point.

(4) Is there also PCP within the Wnt3 expressing domain? In other words, (and linked to question 2), does PCP require a certain concentration of Wnt3 or a gradient of Wnt3 in order to provide an orientation?

In the context of a simple Wn3-MO rescue experiment, PCP is coordinated within the Wnt3-positive area. But this could be because PCP can propagate in both orientations, so it does not answer the question. In the Wnt3-rescue experiments in Fmi-MO and Stbm-MO embryos, PCP seemed better oriented close to the boundary between Wnt3-positive and -negative areas, in particular outside the Wnt3-positive area and rather uncoordinated deep in the middle of Wnt3-RNA positive area.

If Wnt3 expression is uniform across an embryo, as achieved by Wnt3-mRNA injection into the egg, the axis will be lost entirely (Momose 2008). We interpret these observations as indicating that Wnt3 expression "contrasts" (or steep gradients) act as the PCP orientation cue rather than a permissive manner.

In normal development, mRNA expression detected by in situ hybridisation has a slight gradient, but we do not have any information about the endogenous protein distribution.

We greatly appreciate the reviewer’s insightful comments. A few sentences addressing points (2) and (4) have been added. The graphical models in Figures 4 and 6A have been updated. While these are relatively minor changes to the manuscript, they significantly impact future perspectives.

Minor comments:(1) Labeling in some of the figures is too small and not legible, e.g. Figures 4E-H. Please check and improve.

Agreed. Some labelling was way too small (2.5 points). This has been corrected. The minimum font size is now 6-point for most labelling in the revised Figures.

(2) Page 13: ...and allow us to novel scenarios for PCP-driven axis symmetry breaking... seems to lack the verb "propose"

Corrected.